# Stratification of TAD boundaries reveals preferential insulation of super-enhancers by strong boundaries

Yixiao Gong[1,2], Charalampos Lazaris[1,2], Theodore Sakellaropoulos[3], Aurelie Lozano[4], Prabhanjan Kambadur[5], Panagiotis Ntziachristos[6], Iannis Aifantis[1,2] & Aristotelis Tsirigos [1,2,7]

The metazoan genome is compartmentalized in areas of highly interacting chromatin known as topologically associating domains (TADs). TADs are demarcated by boundaries mostly conserved across cell types and even across species. However, a genome-wide characterization of TAD boundary strength in mammals is still lacking. In this study, we first use fused two-dimensional lasso as a machine learning method to improve Hi-C contact matrix reproducibility, and, subsequently, we categorize TAD boundaries based on their insulation score. We demonstrate that higher TAD boundary insulation scores are associated with elevated CTCF levels and that they may differ across cell types. Intriguingly, we observe that super-enhancers are preferentially insulated by strong boundaries. Furthermore, we demonstrate that strong TAD boundaries and super-enhancer elements are frequently co-duplicated in cancer patients. Taken together, our findings suggest that super-enhancers insulated by strong TAD boundaries may be exploited, as a functional unit, by cancer cells to promote oncogenesis.

[1] Department of Pathology, NYU School of Medicine, New York, NY 10016, USA. [2] Laura and Isaac Perlmutter Cancer Center and Helen L. and Martin S. Kimmel Center for Stem Cell Biology, NYU School of Medicine, New York, NY 10016, USA. [3] School of Mechanical Engineering, National Technical University of Athens, Zografou 15780, Greece. [4] Center for Computational and Statistical Learning, IBM T.J. Watson Research Center, New York, NY 10598, USA. [5] Bloomberg LP, 731 Lexington Avenue, New York City, NY 10022, USA. [6] Department of Biochemistry and Molecular Genetics, Feinberg School of Medicine, Northwestern University, Chicago, IL 60611, USA. [7] Applied Bioinformatics Laboratories, NYU School of Medicine, New York, NY 10016, USA. Yixiao Gong and Charalampos Lazaris contributed equally to this work. Correspondence and requests for materials should be addressed to I.A. (email: Ioannis.Aifantis@nyumc.org) or to A.T. (email: Aristotelis.Tsirigos@nyumc.org)

The advent of proximity-based ligation assays has allowed scientists to probe the three-dimensional chromatin organization at an unprecedented resolution[1,2]. Hi-C, a high-throughput chromosome conformation variant, has enabled genome-wide identification of chromatin–chromatin interactions[3]. Hi-C has revealed that the metazoan genome is organized in areas of active and inactive chromatin known as A and B compartments, respectively[3]. These are further compartmentalized into super-TADs[4], topologically associating domains (TADs)[5–7] and sub-TADs[8], as well as gene neighborhoods[9]. Several algorithms have been developed to reveal this hierarchical chromatin organization, including Directionality Index (DI)[5], Armatus[10], TADtree[11], insulation index (Crane)[12], IC-finder[13], and others. However, none of these studies has systematically explored the properties of TAD boundaries. Although TADs are seemingly invariant across cell types, mounting evidence suggests that TAD boundaries can vary in strength, ranging from permissive ("weak") TAD boundaries that allow more inter-TAD interactions to more rigid ("strong") boundaries that clearly demarcate adjacent TADs[14]. Recent studies have shown that in *Drosophila*, exposure to heat-shock caused local changes in certain TAD boundaries resulting in TAD merging[15]. Another recent study showed that during motor neuron (MN) differentiation in mammals, TAD, and sub-TAD boundaries in the *Hox* cluster are not rigid and their plasticity is linked to changes in gene expression during differentiation[16]. It has also been demonstrated that boundary strength is positively associated with the occupancy of structural proteins, including CCCTC-binding factor (CTCF)[5]. Despite these advances, no study has yet addressed the issue of boundary strength in mammals and how it may be related to potential boundary disruptions and aberrant gene activation in cancer. Here we first introduce a new method based on fused two-dimensional (2D) lasso[17] in order to improve Hi-C matrix reproducibility. Then, we use the improved Hi-C matrices to: (a) categorize TAD boundaries based on their insulating strength, (b) characterize TAD boundaries in terms of CTCF binding and other functional elements, and (c) investigate potential genetic alterations of TAD boundaries in cancer. We anticipate that our study will help generate new insights into the significance of TAD boundaries.

## Results

**Analysis workflow.** The overall workflow, including our benchmark strategy and downstream analysis, is summarized in Fig. 1. Initial alignment and filtering of the collected Hi-C sequencing data sets was performed with Hi-C-bench[18] (see Methods section for details). Quality assessment analysis revealed that the samples varied considerably in terms of total numbers of reads, ranging from ~150 million reads to >1.3 billion (Supplementary Figure 1a). Mappable reads were over 96% in all samples. The percentages of total accepted reads corresponding to *cis* (ds-accepted-intra, dark green) and *trans* (ds-accepted-inter, light green) (Supplementary Figure 1b) also varied widely, ranging from ~17 to ~56%. The characteristic drop of average Hi-C signal as a function of distance between interacting loci was observed (Supplementary Figure 1c). The main part of analysis starts with unprocessed Hi-C contact matrices ("filtered" matrices). We then generate processed Hi-C matrices using ICE "correction"[19], our "scaling" approach (Methods section) and calCB[20]. Finally, fused two-dimensional lasso is applied on the processed Hi-C matrices. Matrix reproducibility between biological replicates is assessed across samples for a variety of parameters, for example, resolution, distance between interacting loci, sequencing depth, and so on, using stratum-adjusted correlation coefficients[21]. Finally, downstream analysis, involves the characterization of TAD boundaries based on their insulating strength, the enrichment in CTCF binding, proximity to repeat elements and super-enhancers, and, finally, their genetic alterations in cancer.

**Reproducibility assessment of Hi-C contact matrices.** Hi-C is prone to biases and multiple algorithms have been developed for Hi-C bias correction, including probabilistic modelling methods[22], Poisson or negative binomial normalization[23], calCB which

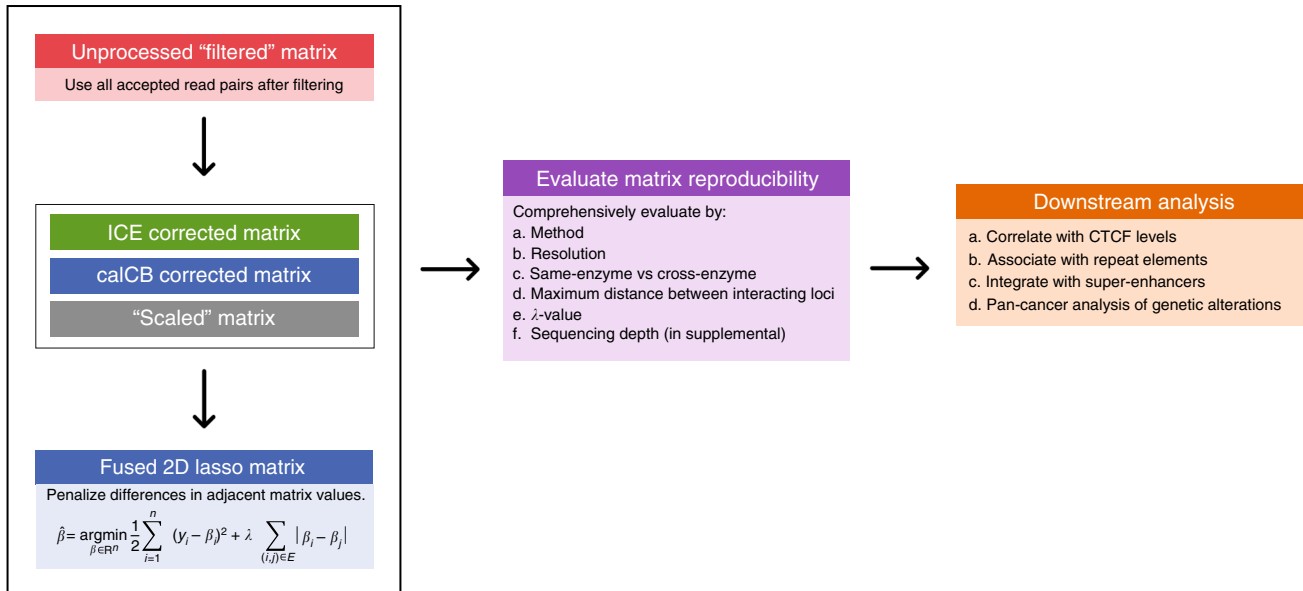

**Fig. 1** Overall workflow and benchmarking strategy. Our analysis starts with unprocessed Hi-C contact matrices. We then generate processed Hi-C matrices using ICE "correction", our "scaling" approach and calCB. Fused two-dimensional lasso is applied on the processed Hi-C matrices. Matrix reproducibility between biological replicates is assessed across samples for a variety of parameters using stratum-adjusted correlation coefficients[21]. Finally, downstream analysis, involves the characterization of TAD boundaries based on their insulating strength, the enrichment in CTCF binding, proximity to repeat elements and super-enhancers, and, their genetic alterations in cancer

corrects for copy number variation (CNV) [20] and the widely used Iterative Correction and Eigenvalue decomposition method (ICE) [19] which assumes "equal visibility" of genomic loci. A similar iterative method named Sequential Component Normalization was introduced by Cournac et al.[24]. Additional efficient correction methods have been developed to handle high-resolution Hi-C data sets[25]. However, estimating highly reproducible Hi-C contact maps remains a challenging task[26], especially at high resolutions, as we also demonstrate below. Specifically, we focused on multiple factors that may play an important role on reproducibility: first, we separately considered biological replicates of Hi-C libraries generated with the same or different restriction enzymes; second, we studied the impact of Hi-C matrix resolution (i.e., bin size); third, we assessed reproducibility as a function of the distance of interacting loci pairs; fourth, we studied the impact of sequencing depth. Stratum-adjusted correlation coefficients (SCC) were calculated for each pair of replicates (same- or cross-enzyme) on Hi-C contact matrices estimated by four methods: (i) naive filtering (i.e., matrix generation by simply using double-sided accepted intra-chromosomal read pairs from Supplementary Figure 1a), (ii) iterative correction (ICE) which has been demonstrated to improve cross-enzyme correlation, (iii) calCB which corrects for known Hi-C biases, as well as for CNV, and (iv) our own scaling method which also corrects for effective length, GC content and mappability (see Supplementary Figure 2a, b and Methods section for details). The results of our benchmark analysis are summarized in Supplementary Figure 3: the left panel summarizes the correlations between replicates generated by the same restriction enzyme, whereas the right panel the correlations between replicates generated by a different restriction enzyme. In both scenarios, as expected, reproducibility drops quickly as finer resolutions (from 100 to 20 kb) are considered. The same conclusion applies for increasing distance (from 2.5 to 10 Mb) between interacting loci, demonstrating that long-range interactions require ultra-deep sequencing (beyond what is currently available in most of the data sets in this study) in order to be detected reliably. To elaborate on this point, we repeated the analysis after resampling at higher sequencing depth (Supplementary Figure 4). Both conclusions hold true with the new sequencing depth and are independent of the Hi-C contact matrix estimation method. From this benchmarking study, we conclude that reproducibility of Hi-C contact matrices across biological replicates is not ideal and that there is a need for computational methods to improve it. In the next sections, we focus on improving the reproducibility of the Hi-C contact matrices within the context of TADs, as most of the DNA-DNA interactions occur within these domains. Since TAD sizes typically range from 200 to 2.5 Mb (>92% of all TADs identified in our Hi-C data sets), and, as demonstrated in Supplementary Figure 3 and Supplementary Figure 4, stratum-adjusted correlation coefficients between biological replicates of Hi-C contact matrices drop dramatically beyond 2.5 Mb, we restrict our subsequent analyses to distances up to 2.5 Mb.

**Fused lasso improves reproducibility of Hi-C matrices**. Motivated by the poor performance of all methods at fine resolutions and by the observation of a trade-off between cross-enzyme and same-enzyme reproducibility when correcting for enzyme-related biases, we decided to utilize a machine learning denoising method, fused 2D lasso[27], to improve the reproducibility of Hi-C contact matrices. Briefly, 2D fused lasso introduces a parameter $\lambda$ which penalizes differences between neighboring values in the Hi-C contact matrix (Methods section for details). The effect of parameter $\lambda$ is demonstrated in Fig. 2a where we show an

example of the application of fused 2D lasso on a Hi-C contact matrix focused on an 8 Mb locus on chromosome 8 (chr8:124700000–132700000) for different values of parameter $\lambda$. To evaluate the performance of fused lasso, we calculated same-enzyme and cross-enzyme stratum-adjusted correlation coefficient (SCC) values between Hi-C contact matrices generated from different replicates. SCC values were calculated either for iteratively corrected (ICE), calCB-corrected or scaled Hi-C contact matrices (at different $\lambda$ values) and compared to the naïve filtering approach. The results for same enzyme, are summarized in Fig. 2b. Increasing $\lambda$ improves reproducibility independent of resolution, restriction enzyme, and bias-correction method, demonstrating the robustness of our approach. Similarly, fused 2D lasso improves the reproducibility of contact matrices in the cross-enzyme case, as demonstrated in Fig. 2c. The same analysis was performed at lower sequencing depth with similar results (Supplementary Figure 5). Next, we explored the effect of fused 2D lasso on Hi-C matrices of fine resolutions. For this analysis, we used 5 kb bins to compute the interaction matrix. To compensate for distance-related biases in Hi-C matrices (Supplementary Figure 1c), we normalized the interaction strength for every distance/diagonal using a robust version of $z$-score (see Methods section for details). Then, we applied the Graph-Fused Lasso implementation of fused 2D lasso[28], which scales better than the Fused Lasso Signal Approximator (flsa)[29] used for coarse resolutions. Since available Hi-C data sets lack biological replicates of ultra-deep sequenced samples, we evaluated our method by testing whether it could recover the 5 kb loops identified in Rao et al.[30] in a single-biological sample of GM12878, the most deeply sequenced sample in this study (~3 billion read pairs of which ~900 million intra-chromosomal read pairs passed our filtering criteria). As a recovery metric, we used the fraction of the reported loops within the top interactions as ranked by our fused lasso approach. We observed that by tuning the $\lambda$ parameter we improved this metric by an 8% relative improvement (Supplementary Figure 6a). For the optimal $\lambda$, our method ranked most of the known loops (~90%) in the top 10% of measured interactions (~79% in the top 5% of all measured interactions). We also evaluated the sensitivity of our approach to subsampling. In particular, we re-computed the interaction matrices using 200, 400, 600, 800 million intra-chromosomal read pairs, re-run the analysis and obtained relative improvements of 9%, 14%, 22%, and 32%, respectively (Supplementary Figure 6a). Performance of the graph-fused lasso algorithm as a function of chromosome size is presented in Supplementary Figure 6b (peak memory consumption) and Supplementary Figure 6c (execution time).

**Fused lasso preserves cell-type specificity of Hi-C matrices**. Although fused 2D lasso improves reproducibility of Hi-C matrices between biological replicates, there is a possibility that this is achieved at the expense of losing cell-type specificity. To test this, we compared the effect of $\lambda$ on the reproducibility between biological replicates (intra-cell-type) to its effect on the stratum-adjusted correlation coefficients between unrelated samples in our Hi-C data set collection (inter-cell-type). For this test, we chose to focus on the collection of H1 stem cell Hi-C replicates and their derivatives generated by the Ren lab[31], so that we could assess the effect of smoothing on subtle cell-type specific differences in experiments performed in a single lab. Hi-C matrices were distance-normalized (similar to Yan et al.[32], see Methods section for details) to account for the dependence of the Hi-C signal on the distance between interacting loci. The results of this analysis are presented in Fig. 3: although both intra-cell-type and inter-cell-type stratum-adjusted correlation coefficients

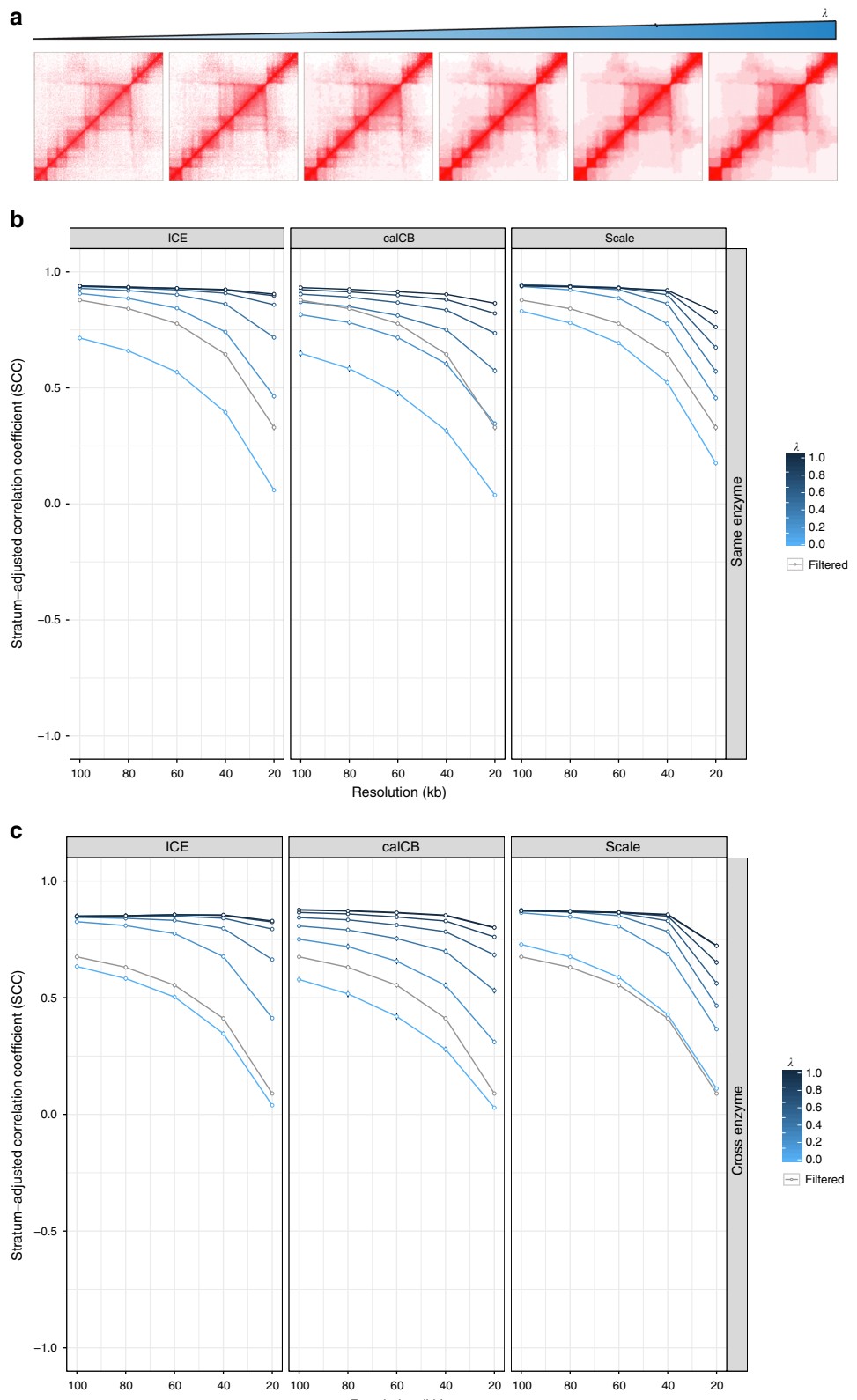

**Fig. 2** Fused two-dimensional lasso improves reproducibility of Hi-C contact matrices (high sequencing depth = 80 million intrachromosomal read pairs). **a** Example of application of fused two-dimensional lasso on a Hi-C contact matrix focused on a 8 Mb locus on chromosome 8 for different values of parameter $\lambda$. **b** Stratum-adjusted correlation coefficient values are improved by increasing the value of fused lasso parameter $\lambda$ for matrices estimated with ICE, calCB and our simple scaling method (same enzyme). **c** Stratum-adjusted correlation coefficient values are improved by increasing the value of fused lasso parameter $\lambda$ for matrices estimated with ICE, calCB, and our simple scaling method (cross enzyme). As a baseline control, stratum-adjusted correlation coefficients of Hi-C contact matrices generated by the naive filtering method are marked by the gray line in each panel

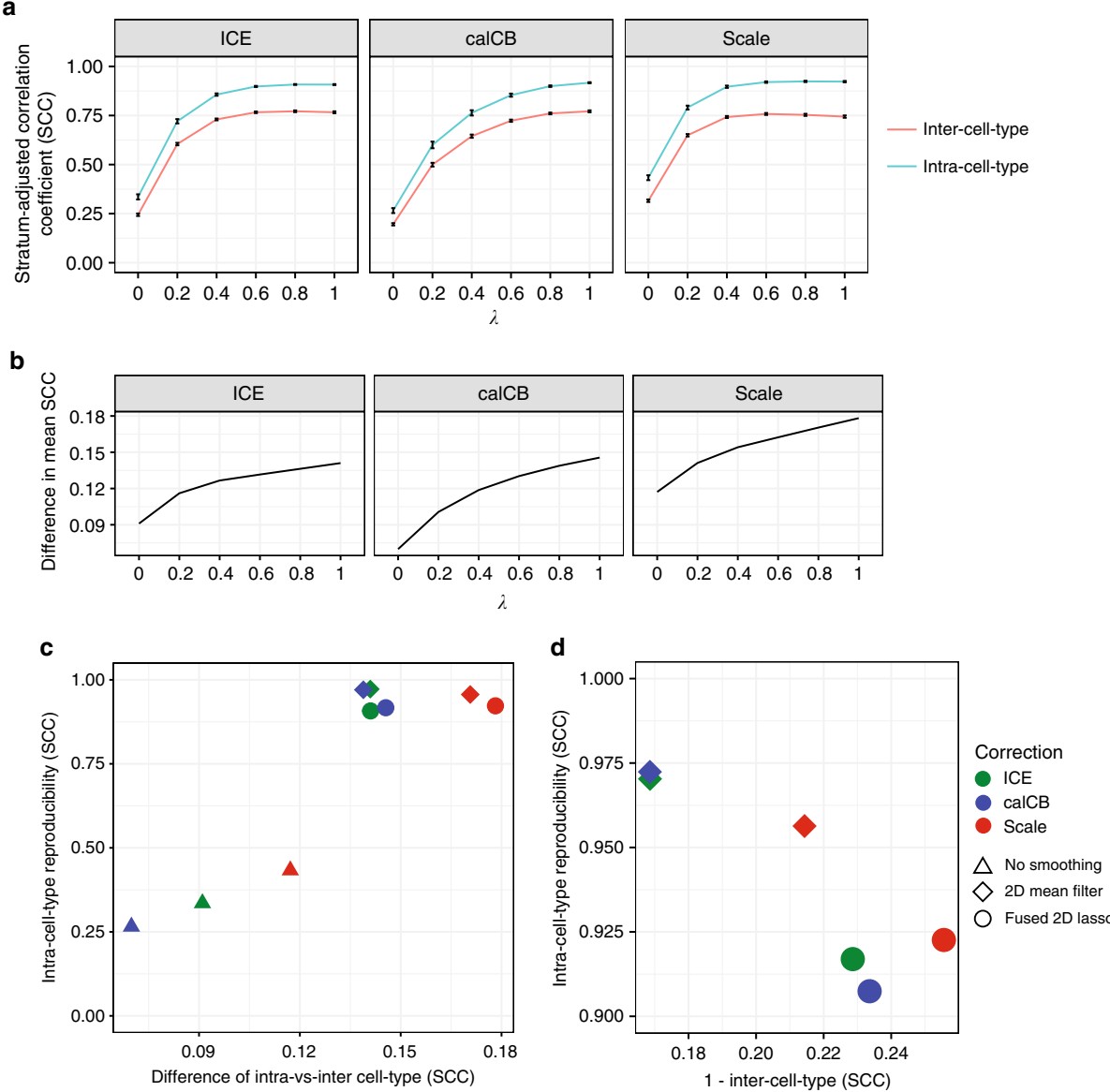

**Fig. 3** Fused lasso preserves cell-type specificity of Hi-C contact matrices (high sequencing depth = 80 million intrachromosomal read pairs). **a** Effect of fused lasso on stratum-adjusted correlation coefficient for the case of intra-cell-type (cyan) and inter-cell-type (orange) comparisons. Matrices of 40 kb resolution were used for the analysis. The Hi-C matrices were processed with ICE, calCB, or scaling matrix correction methods. **b** Difference in mean stratum-adjusted correlation coefficient between intra-cell-type and inter-cell-type sample comparisons. **c** Comparison of Hi-C matrix "correction" and smoothing methods in terms of preservation of cell-type specificity (x-axis) and intra-cell-type reproducibility (y-axis). **d** Comparison of Hi-C matrix "correction" and smoothing methods in terms of (1—inter-cell-type) (x-axis) and intra-cell-type (y-axis) stratum-adjusted correlation coefficients

increase by $\lambda$ (Fig. 3a), the difference between intra-cell-type and inter-cell-type correlation coefficients also increases (Fig. 3b), suggesting that fused 2D lasso actually preserves cell-type specificity of Hi-C contact matrices, a behavior that is consistent independent of the matrix "correction" method. Nevertheless, some "correction" methods appear to work better than others in combination with lasso. In addition, we also evaluated an alternative "smoothing" method, 2D mean filter smoothing, recently made available as part of the HiCRep package[21]. In Fig. 3c, we show the results of the comparison of the three correction methods in combination with the smoothing techniques using two metrics: preservation of cell-type specificity (x-axis) and intra-cell-type reproducibility (y-axis). The main conclusions from this comparison are: (a) smoothing (lasso or mean filter) improves both metrics independent of the correction method, and (b) fused lasso performs slightly better than mean filter

smoothing in preserving cell-type specificity, while it behaves slightly worse in improving intra-cell-type specificity. In Fig. 3d, we further demonstrate the trade-off between intra-cell-type and inter-cell-type metrics when using 2D lasso or 2D mean filtering.

**Fused lasso reveals a nested TAD hierarchy**. After demonstrating that parameter $\lambda$ improves reproducibility of Hi-C contact matrices independent of the bias-correction method, we hypothesized that increased values of $\lambda$ may also define distinct classes of TADs with different properties. For this reason, we now allowed $\lambda$ to range from 0 to 5. We then identified TADs at multiple $\lambda$ values using Hi-C-bench on Hi-C matrices binned at 40 kb (all downstream analyses rely on TAD calling performed on Hi-C matrices at 40 kb), and we observed that the number of TADs is monotonically decreasing with the value of $\lambda$

(Supplementary Figure 7a), suggesting that by increasing $\lambda$, we are effectively identifying larger TADs encompassing smaller TADs detected at lower $\lambda$-values. Indeed, when comparing TAD boundaries detected at successive $\lambda$ values, we found that higher $\lambda$-values produced TAD boundaries that are almost a strict subset of TAD boundaries produced at lower $\lambda$ values (~94% overlap when considering only the exact bin as a true common TAD boundary, and ~98% when TAD boundaries are allowed to differ by at most one bin between TADs generated for successive $\lambda$-values). Equivalently, certain TAD boundaries "disappear" as $\lambda$ is increased. Therefore, we hypothesized that TAD boundaries that disappear at lower values of $\lambda$ are weaker (i.e., lower insulation score), whereas boundaries that disappear at higher values of $\lambda$ are stronger (i.e., higher insulation score). To test this hypothesis, we identified the TAD boundaries that are "lost" at each value of $\lambda$, and generated the distributions of the insulation scores for each $\lambda$ across samples. As insulation score, we used the Hi-C "ratio" score (Methods section), which was shown to outperform other TAD calling methods[18]. Indeed, as hypothesized, TAD boundaries lost at higher values of parameter $\lambda$ are associated with higher TAD insulation scores (Supplementary Figure 7b).

**Stratification of TAD boundaries by insulating score**. Motivated by the observation that with increasing $\lambda$, weaker TAD boundaries are not detected, we decided to explore in depth the properties of TAD boundaries with respect to their insulation score. To this end, we stratified TAD boundaries into five categories (I through V) of equal size according to their insulation score, independently in each Hi-C data set used in this study. As shown in Fig. 4a, we first processed the Hi-C matrices using ICE, calCB and scaling and applied fused 2D lasso with "optimal" $\lambda$, defined as the $\lambda$ value beyond which no statistically significant improvement on the reproducibility is observed. The statistical significance was assessed using a Wilcoxon test between the distributions of stratum-adjusted correlation coefficients across chromosomes in given sample for successive $\lambda$ values. The procedure is demonstrated using an IMR90 replicate as an example (Supplementary Figure 7c). Then, TAD calling and TAD boundary insulation score calculations were performed using our "ratio" method (see Methods section for details) and the boundaries were classified into five equal-size categories, as mentioned above. A heatmap representation, including all TAD boundaries and their associated boundary strength category across all samples is depicted in Fig. 4b ("NA" corresponds to lack of boundary, as it is possible that boundaries called in certain samples are not present in others). Unbiased hierarchical clustering correctly grouped replicates and related cell types independent of enzyme biases or batch effects related to the lab that generated the Hi-C libraries, suggesting that TAD boundary strength can be used to distinguish cell types. Equivalently, this finding suggests that, although TAD boundaries have been shown to be largely invariant across cell types, a certain subset of TAD boundaries may exhibit varying degrees of strength in different cell types. Also, as expected, TAD boundary strength was found to be positively associated with CTCF levels, suggesting that stronger CTCF binding confers stronger insulation. Since we noticed that several TAD boundaries contain transcriptional start sites (TSSs), this analysis was done separately for TSS-only CTCF peaks (Fig. 4c) and for all CTCF peaks (see below). Both approaches revealed the same trend, with the exception of the class of strongest boundaries (category V), where CTCF levels in TSS regions were significantly higher compared to non-TSS regions, suggesting that the strongest boundaries are formed by CTCF-mediated loops at gene promoters. Alternatively to our "ratio" insulation score, we repeated our analysis using the insulation

score generated by the "crane" TAD calling algorithm[12]. A comparative analysis with between "ratio" and "crane" is shown in Fig. 4d, where it appears that ratio-generated insulation scores better associate with CTCF levels. In the interest of robustness, we performed the same analyses for all preprocessing methods, at both low and high sequencing depth, for both "ratio" and "crane" insulation scores (Supplementary Figure 8 and Supplementary Figure 9, respectively), for TSS-only CTCF peaks (Supplementary Figure 8a and Supplementary Figure 9a), as well as for all CTCF peaks (Supplementary Figure 8b and Supplementary Figure 9b). Finally, SINE elements have also been shown to be enriched at TAD boundaries[5], and besides confirming this finding, we now demonstrate that Alu elements (the most abundant type of SINE elements) are enriched at stronger TAD boundaries (Supplementary Figure 10, top-left panel). A comprehensive analysis of all major repetitive element subtypes (downloaded from the UCSC Genome Browser[33]) can be found in Supplementary Figure 10.

**Super-enhancers are insulated by strong TAD boundaries**. We then explored what type of functional elements are localized within TADs demarcated by strong TAD boundaries. Specifically, we tested super-enhancers identified in matched samples (see Methods section for details). Super-enhancers are key regulatory elements thought to be defining cell identity[9,34], and are usually found near the center of TADs[35]. Our analysis determined that they are significantly more frequently localized within TADs insulated by at least one strong TAD boundary (Fig. 4e). Further analysis revealed that, super-enhancers are 2.94 times more likely to be insulated by strong boundaries (categories IV or V) in both the upstream and downstream directions, compared to being insulated by weak boundaries (categories I or II) in both directions. A comparison with TAD boundary classification using "crane" insulation scores demonstrated that "ratio" insulation scores are more significantly associated with proximity to super-enhancers (Fig. 4f). A similar robustness analysis as the one presented above for CTCF was also performed for super-enhancers (Supplementary Figure 8c and Supplementary Figure 9c for "ratio" and "crane" insulation scores, respectively). Taken together, our findings suggest that, because of their significance in gene regulation, super-enhancers should only target genes confined in the "correct" TAD or neighborhood, while remaining strongly insulated from genes in adjacent TADs. This is conceivably achieved by the strong TAD boundaries we have identified in this study.

**Strong TAD boundaries are co-duplicated with super-enhancers**. To further investigate the importance of variable boundary strength, we asked whether TAD boundaries are prone to genetic alterations in cancer. To this end, we mined structural variants released by the International Cancer Genome Consortium (ICGC)[36]. A summary of the reported variant types across all cancer types available on ICGC, is presented in Supplementary Figure 11. First, for each focal (up to 1 Mb) deletion event, we identified the TAD boundaries closest to the breakpoints, and calculated the frequency of deletions by boundary strength. We observed that the frequency of deletions monotonically decreased with increasing boundary strength (Fig. 5a). This suggests that strong TAD boundaries are less frequently lost in cancer, as they may "safeguard" functional elements that are necessary for proliferation. By contrast, the frequency of tandem duplications (up to 1 Mb) increased with increasing boundary strength (Fig. 5b). Both results were robust to various cutoffs on the sizes of the structural variants, within the usual range of TAD sizes (from 250 kb to 2.5 Mb). Then, to

further clarify the connection between super-enhancers, strong TAD boundaries and cancer, we studied tandem duplication events where super-enhancers (obtained from a publicly available collection of super-enhancers[37]) are co-duplicated with adjacent strong boundaries. As demonstrated in Fig. 5c, super-enhancers are indeed co-duplicated with strong TAD boundaries.

Co-duplication of strong boundaries and super-enhancers was statistically significantly more frequent than that of strong boundaries and regular enhancers. This suggests that, in cancer, not only are strong boundaries protected from deletions, but they are also co-duplicated with super-enhancer elements. A robustness analysis similar to the one performed for CTCF and super-

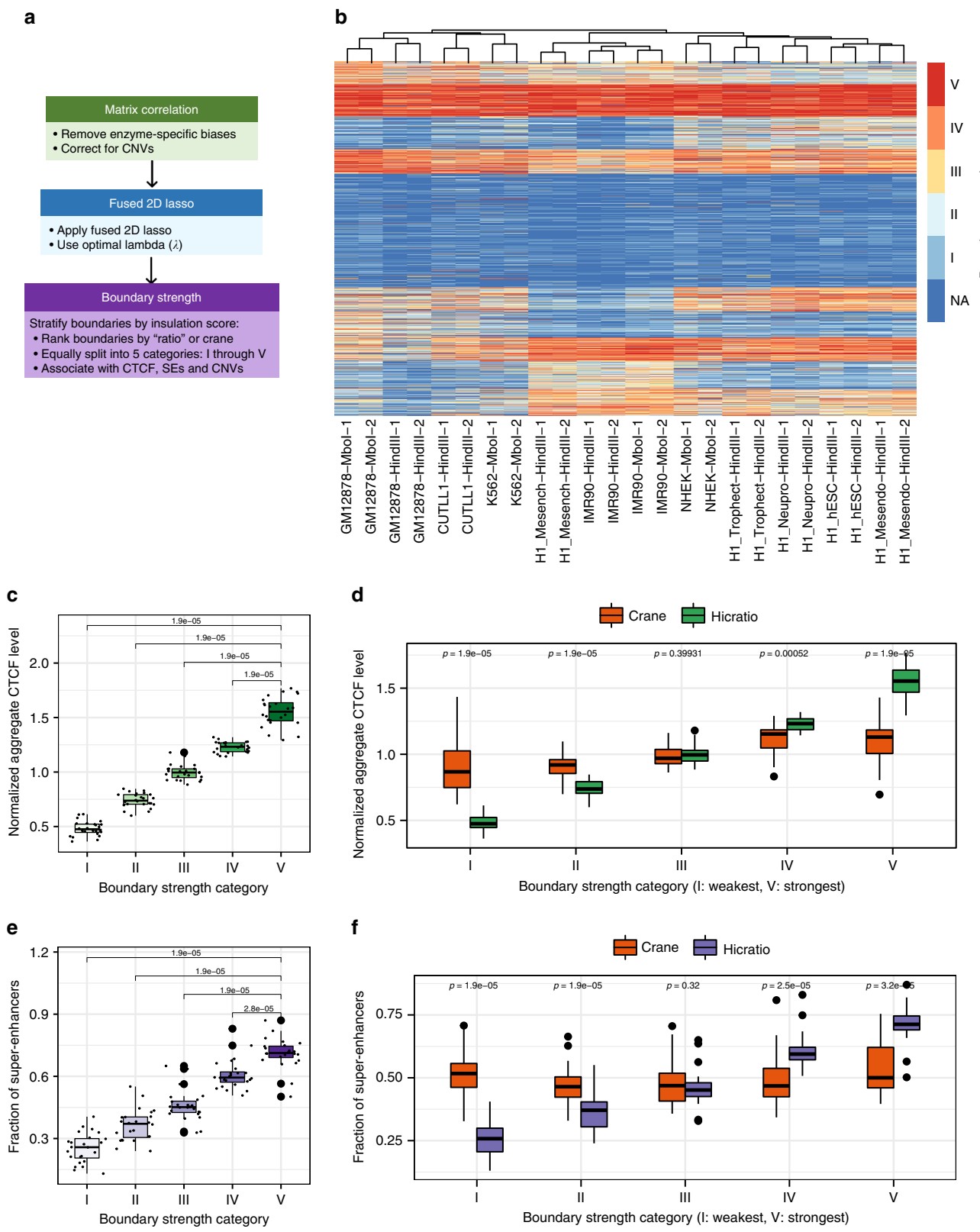

enhancers is presented in Supplementary Figure 12 demonstrating that our findings are consistent for low and high sequencing depth. Finally, we present an example of a co-duplication of a super-enhancer with a strong boundary in Fig. 5d: *MYC*, a well-known oncogene that is typically overexpressed in cancer, is localized next to a strong TAD boundary and is co-duplicated with the boundary, as well as with several proximal super-enhancers.

## Discussion

Multiple recent studies have revealed that the metazoan genome is compartmentalized in boundary-demarcated functional units known as topologically associating domains (TADs). TADs are highly conserved across species and cell types. A few studies, however, provide compelling evidence that specific TADs, despite the fact that they are largely invariant, exhibit some plasticity. Given that TAD boundary disruption has been recently linked to aberrant gene activation and multiple disorders including developmental defects and cancer, categorization of boundaries based on their strength and identification of their unique features becomes of particular importance. In this study, we first developed a method based on fused 2D lasso in order to improve Hi-C contact matrix reproducibility between biological replicates. Then, we categorized TAD boundaries based on their insulating score. Our analysis demonstrated that: (a) using fused 2D lasso, we can improve the reproducibility of Hi-C contact matrices irrespective of the Hi-C bias correction method used, and (b) using our "ratio" insulation score, we can successfully identify boundaries of variable strength and that strong boundaries exhibit certain expected features, such as elevated CTCF levels. By performing an integrative analysis of boundary strength with super-enhancers in matched samples, we observed that super-enhancers are preferentially insulated by strong boundaries, suggesting that super-enhancers and strong boundaries may represent a biologically relevant entity. Motivated by this observation, we examined the frequency of structural alterations involving strong boundaries and super-enhancers. We found that not only strong boundaries are "protected" from deletions, but, more importantly, they are co-duplicated together with super-enhancers. Recently, it has been shown that genetic or epigenetic alterations near enhancers may lead to aberrant activation of oncogenes[38–41]. Our results, expand on these studies by highlighting a previously unknown connection between strong TAD boundaries, super-enhancers and tandem duplication events in cancer.

## Materials

**Processing of published high-resolution Hi-C data sets**. In order to develop and benchmark a method that improves reproducibility of Hi-C contact matrices, we carefully selected our Hi-C data sets to represent technical variation due to the execution of the experiments by different laboratories and/or the usage of different restriction enzymes. We identified publicly available human Hi-C data sets that fulfilled the following criteria: (i) availability of two biological replicates and (ii) sufficient

sequencing depth to robustly identify topologically associating domains (TADs) as described in our TAD calling benchmark study[18]. Specifically, we ensured that our data sets included samples with at least ~40 million intra-chromosomal read pairs and that the Hi-C experiment was performed in biological replicates, either by using one restriction enzyme (HindIII or MboI) (H1 cells and their derivatives[31], K562, KBM7, and NHEK cells[30] and in-house generated CUTLL1), or two enzymes (HindIII or MboI) (GM12878[30], IMR90[5,42]), in order to examine the consistency of predicted Hi-C interactions across different enzymes. Detailed information about the Hi-C data sets, including cell type and GEO accession number, is listed in Supplementary Data 1. All data sets were then comprehensively re-analysed using our Hi-C-bench platform[18]. Briefly, paired-end reads were mapped to the reference genome (hg19) using Bowtie2[43]. Reads with low mapping quality (MAPQ <30) were discarded. Local alignment of input read pairs was performed, as they often consist of chimeric reads between two (non-consecutive) interacting fragments. Mapped read pairs were subsequently filtered for known artifacts of the Hi-C protocol, such as self-ligation, mapping too far from the enzyme's known cutting sites, etc, using GenomicTools[44] gtools-hic filter command. More specifically, reads mapping in multiple locations on the reference genome (multihit), double-sided reads that mapped to the same enzyme fragment (ds-same-fragment), reads whose 5′-end mapped too far (ds-too-far) from the enzyme cutting site, reads with only one mappable end (single sided) and unmapped reads (unmapped), were discarded. Read pairs that corresponded to regions that were very close (<25 kilobases, ds-too-close) in linear distance on the genome as well as duplicate read pairs (ds-duplicate-intra and ds-duplicate-inter) were also discarded. Quality assessment analysis revealed that the samples varied considerably in terms of total numbers of reads, ranging from ~150 million reads to >1.3 billion. Mappable reads were over 96% in all samples. The percentages of total accepted reads corresponding to *cis* (ds-accepted-intra) and *trans* (ds-accepted-inter) also varied widely, ranging from ~17 to ~56%. Despite the differences in sequencing depth and in the percentages of useful reads across samples, all samples had enough useful reads for TAD detection. Due to the wide differences in sequencing depth, and to ensure fair comparisons of Hi-C matrices in this study, all data sets were down-sampled such that the number of usable intra-chromosomal reads pairs was ~40 million for each replicate. To study the effect of sequencing depth, we also resampled at ~80 million usable intra-chromosomal read pairs. Finally, Hi-C contact matrices were generated using fixed bin sizes at multiple resolutions (5, 20, 40, 60, 80, and 100 kb).

**Scaled Hi-C contact matrices**. Hi-C contact matrices were scaled by: (a) the total number of (usable) intra-chromosomal read pairs, and (b) the "effective length" of the corresponding pair of interacting bins. The effective length of a genomic bin was previously defined as the total length of genomic regions that fall within a specified distance (typically 500 nt) from a restriction

**Fig. 4** Classification and characterization of TAD boundaries according to insulation score (high sequencing depth = 80 million intrachromosomal read pairs). **a** Workflow of stratification of TAD boundaries by insulating score. **b** Heatmap representation of TAD boundary insulation strength across samples; hierarchical clustering correctly groups replicates and related cell types independent of enzyme biases or batch effects related to the lab that generated the Hi-C libraries (detailed information about all Hi-C data sets and their cell types is included in Supplementary Data 1). **c** TAD boundary strength is associated with CTCF levels. **d** Comparison of the association of "ratio" vs. "crane" insulation scores with CTCF levels. **e** Fraction of super-enhancer elements in the vicinity of boundaries of variable strength. **f** Comparison of the association of "ratio" vs. "crane" insulation scores with respect to proximity to super-enhancers. All statistical tests are paired two-sided Wilcoxon rank-sum tests between distributions defined across samples (each sample is a dot in the boxplots). The box in each boxplot represents the first (Q1) and third (Q3) quartiles and the ends of the whiskers are positioned 1.5*(Q3-Q1) away from the ends of the box.

enzyme cutting site[22]. In this study, we defined the scaled Hi-C count corresponding to interactions between the Hi-C matrix bins $i,j$ ($y_{ij}$) as follows:

$$y_{ij} = \frac{x_{ij}}{\text{eff}_i \cdot \text{eff}_j \cdot N},$$

where $x_{ij}$ is the original number of interactions between the bins $i$ and $j$, $\text{eff}_i$ the effective length for the bin $i$, $\text{eff}_j$ the effective length for the bin j, and $N$ is the total number of read pairs. For each bin, at each resolution, effective length, GC content and mappability were calculated as described in Hu et al.[23]. In this study, it was demonstrated that the main source of enzyme-specific biases is

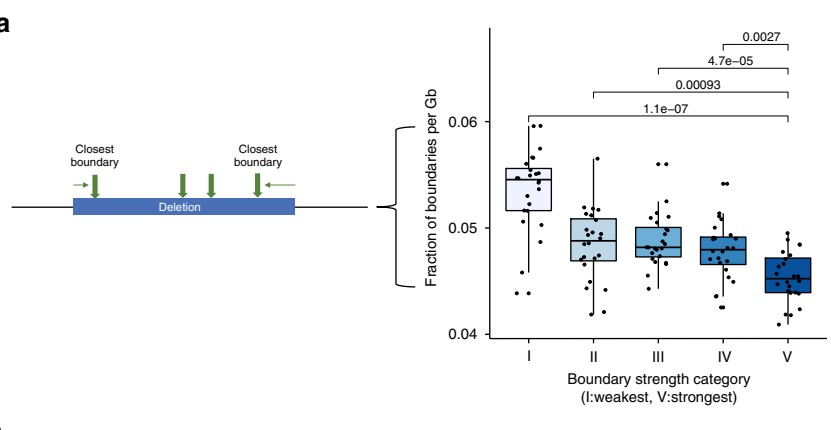

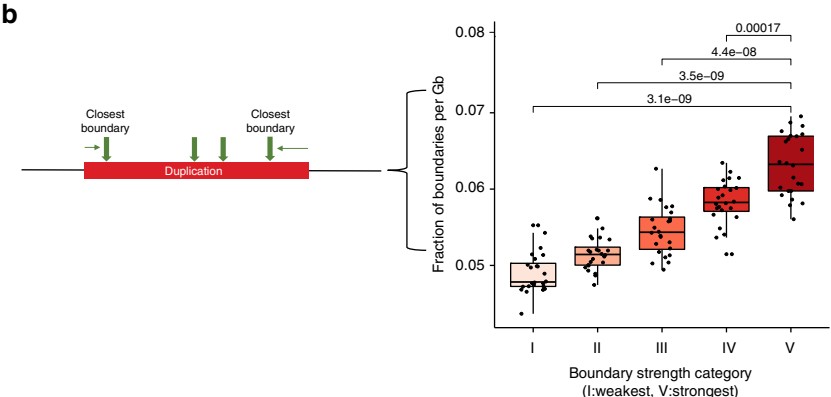

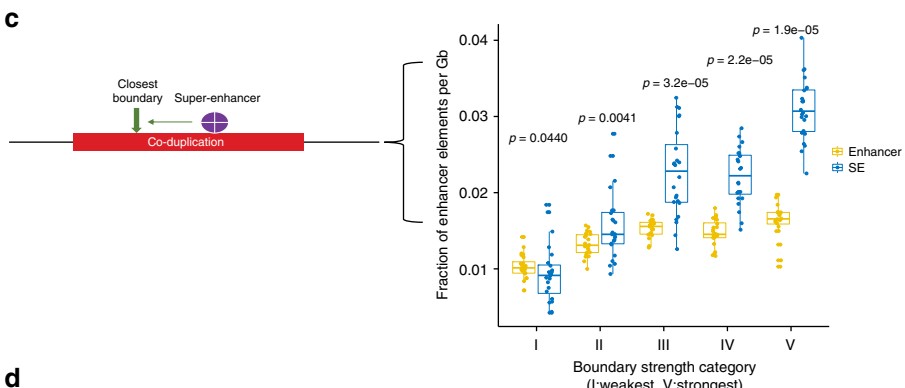

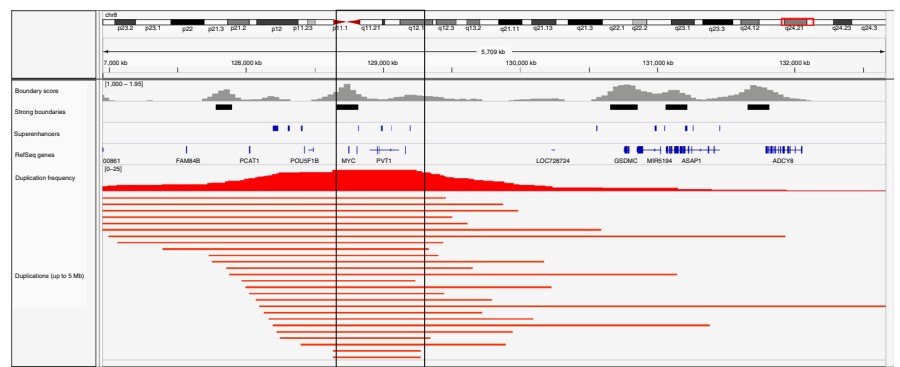

the effective length. Consequently, we expected that correcting for effective length alone would simultaneously correct for GC content and mappability biases. To verify this, we generated heatmaps showing the association of Hi-C interactions with effective length, GC content and mappability[20].

**Distance-normalized Hi-C contact matrices**. Genomic loci that are further apart in terms of linear distance on DNA tend to give fewer interactions in Hi-C maps than loci that are closer. For intra-chromosomal interactions, this effect of genomic distance should be taken into account. Consequently, the interactions were distance-normalized using an adjusted z-score that was calculated taking into account the mean Hi-C counts for all interactions at a given distance $d$ and the corresponding standard deviation. Thus, the z-score for the interaction between the Hi-C contact matrix bins $i$ and $j$ ($z_{ij}$) is given the following equation:

$$z_{ij} = \frac{y_{ij} - m(d)}{mad(d)},$$

where $y_{ij}$ corresponds to the number of interactions between the bins $i$ and $j$, $m(d)$ to the median number of interactions for distance $d=|j-i|$ and $mad(d)$ is the robust estimator of the standard deviation of the mean.

**Fused 2D lasso**. We utilized 2D fused lasso[27], an optimization machine learning technique widely used to analyse noisy data sets, especially images[17]. 2D lasso achieves denoising by penalizing differences between neighboring elements in the contact matrix via a penalty parameter $\lambda$ (lambda), as described in the equation:

$$\hat{\beta} = argmin_{\beta \in \mathbb{R}} n \frac{1}{2} \sum_{i=1}^{n} (y_i - \beta_i)^2 + \lambda \sum_{(i,j) \in E} \left| \beta_i - \beta_j \right|,$$

where $y$ is the original (i.e., observed) contact matrix, and $\hat{\beta}$ is the optimized contact matrix such that the objective function described above in minimized. $E$ describes the neighboring elements of the matrix, i.e., $E = \{ (i, j)$, where $i$ and $j$ are adjacent elements in matrix $\beta\}$.

**Fused 2D lasso packages**. We used two R packages that implement fused 2D lasso:

- the flsa R package (https://cran.r-project.org/web/packages/flsa/index.html), for coarse resolutions (up to 20 kb)[29]
- for fine resolutions, the more recent and more efficient graph-fused lasso python/C++ package (https://github.com/tansey/gfl)[28].

**Calculation of Hi-C matrix reproducibility**. We calculated two types of correlation for Hi-C matrices, to evaluate the performance of our method: (a) same-enzyme reproducibility between Hi-C replicates prepared with the same restriction enzyme, (b) cross-enzyme reproducibility between Hi-C replicates prepared with two different enzymes (e.g., HindIII/MboI). Hi-C matrix reproducibility was assessed using stratum-adjusted correlation coefficient[21] values, calculated on the filtered, ICE-corrected[19], calCB-corrected[20], and scaled Hi-C contact matrices. The ICE and calCB tools have been incorporated into Hi-C-bench (see "Code availability" section), and in this study, they were used with default parameters.

**TAD boundary "ratio" insulation score**. Given a potential TAD boundary, we denote the "upstream" region to the left of the boundary as $L$, and the "downstream" region to the right as $R$. The between regions $L$ and $R$ are denoted as $X$. The "ratio" insulation score is defined as follows:

$$ratio = intra_{max}/inter,$$

where:

$$intra_{max} = max(mean(L), \; mean(R)) \; and \; inter = mean(X).$$

For more details, see Lazaris et al.[18].

**TAD calling using the "ratio" insulation score**. For TAD calling, we first calculated the "ratio" insulation score for each bin at 40 kb resolution. Then, TAD boundaries (of size equal to the bin size, i.e. 40 kb) were identified as local maxima of the insulation scores across each chromosome. Only insulation scores above a certain cutoff were considered as potential TAD boundaries. The cutoff was determined such that the false discovery rate (FDR) of the identified local maxima was not >10%. The FDR was estimated by applying the same procedure (calculate "ratio" insulation scores and seeking local maxima) on randomized Hi-C matrices. The randomized Hi-C matrices were generated by permuting the original matrix values separately for each "diagonal" of the matrix (i.e., Hi-C interaction values at a given distance between interacting loci), so that the distribution of Hi-C signal as a function of distance between interacting loci was preserved in the randomized matrix. The code is publicly available as part of the Hi-C-bench distribution.

**TAD boundary categorization via fused 2D lasso**. We applied 2D fused lasso to categorize TAD boundaries based on their strength. The rationale behind this categorization is that topological domains separated by more "permissive" (i.e., weaker) boundaries[45] will tend to fuse into larger domains when lasso is applied, compared to TADs separated by well-defined, stronger boundaries. We indeed applied this strategy and categorized boundaries into multiple groups ranging from the most permissive to the strongest boundaries. The boundaries that were lost when $\lambda$-value was increased from 0 to 0.25, fall in the first category ($\lambda = 0$), the ones lost when $\lambda$ was increased to 0.5, in the second ($\lambda = 0.25$), and so on.

**Fig. 5** Pan-cancer analysis of strong vs. weak TAD boundaries (high sequencing depth = 80 million intrachromosomal read pairs). **a** Schematic of pan-cancer analysis (left panel) and classification of focally deleted boundaries in cancer according to their strength (right panel). **b** Schematic of pan-cancer analysis (left panel) and classification of focally duplicated boundaries in cancer according to their strength (right panel). **c** Schematic of pan-cancer analysis (left panel) and co-duplications of TAD boundaries with regular enhancers and super-enhancers in cancer (right panel). **d** Snapshot of the *MYC* locus: a strong boundary (black bar) is frequently co-duplicated with *MYC* and potential super-enhancers in cancer patients (highlighted area). IGV tracks from top to bottom: average insulation score across cell types (gray), strong boundaries (black bars), super-enhancer track from SEA (blue bars), RefSeq genes, duplication frequency (red graph) and ICGC patient tandem duplications (red bars). All statistical tests are paired two-sided Wilcoxon rank-sum tests between distributions defined across samples (each sample is a dot in the boxplots). The box in each boxplot represents the first (Q1) and third (Q3) quartiles and the ends of the whiskers are positioned 1.5*(Q3-Q1) away from the ends of the box.

**TAD boundary categorization by insulation score**. We stratified TAD boundaries into five categories (I through V) of equal size according to their insulation score, independently in each Hi-C data set used in this study. Category I contained TAD boundaries with the lowest insulation scores and category V contained those with the highest. Before calculating insulation scores, we first processed the Hi-C matrices using ICE, calCB and scaling and then applied fused 2D lasso (with optimal $\lambda$). Then, TAD calling and TAD boundary insulation score calculations were performed using our "ratio" or the "crane" method and the boundaries were classified into five equal-size categories, as described above.

**Selection of optimal $\lambda$**. For any given Hi-C sample, we defined the "optimal" $\lambda$, as the $\lambda$-value beyond which no statistically significant improvement on the reproducibility is observed. The statistical significance was assessed using a Wilcoxon test between the distributions of stratum-adjusted correlation coefficients (SCC) across chromosomes in given sample for successive $\lambda$ values. Alternatively, the test of statistically significant improvement can be applied on the difference between the intra- and inter-cell-type SCC values in order to take into account similarity to unrelated samples. At least two biological replicates are required for the selection of optimal $\lambda$.

**Analysis of CTCF and H3K27ac ChIP-seq data**. All ChIP-seq data were uniformly processed using the HiC-bench platform[18]. Raw sequencing files were aligned using Bowtie2 version 2.3.1 with standard parameters. Only uniquely mapped reads were selected for downstream analysis. PCR duplicates were removed using Picard-tools version 1.88. MACS version 2.0.10.20131216 were used to call narrow peaks for CTCF and broad peaks for H3K27ac with default parameters.

**Association of CTCF levels with boundary strength categories**. We obtained CTCF ChIP-sequencing data for the cell lines utilized in this study (with the exception of KBM7 for which no publicly available data set was available, see Supplementary Data 1 for details). Total CTCF levels (i.e., aggregated peak intensities from potentially multiple CTCF peaks) at each TAD boundary were calculated and their normalized distributions for each boundary category (weak to strong) were plotted in boxplots in order to demonstrate the association of increased boundary strength with increased levels of CTCF binding. We performed this analysis separately for TSS-only and non-TSS CTCF binding sites. The rationale behind these separate analyses was based on the observation that several TAD boundaries, especially strong boundaries, contain TSSs. Statistical significance was assessed using paired two-sided Wilcoxon rank-sum test. The boxplots represent the distribution of values (normalized CTCF levels) across the Hi-C samples used in this study to define the five categories of TAD boundaries. Detailed information about the CTCF ChIP-seq data sets, including cell type and GEO accession number, is made available in Supplementary Data 1.

**Boundary strength and proximity to super-enhancers**. Super-enhancers were called using H3K27ac ChIP-seq data from GEO, ENCODE and in-house generated data. Detailed information about the H3K27ac ChIP-seq data sets, including cell type and GEO accession number, is made available in Supplementary Data 1. Reads were first aligned with Bowtie2 v2.3.1[43] and then HOMER v4.6[46] was used to call super-enhancers, all with standard parameters. For each super-enhancer in each sample, we identified the corresponding TAD and its TAD boundaries. We then calculated (per sample) the percentage of super-enhancers that are surrounded by boundaries belonging in each boundary

category. Statistical significance was assessed using paired two-sided Wilcoxon rank-sum test. The boxplots represent the distribution of values (fraction of super-enhancers in proximity to TAD boundary categories I through V) across the Hi-C samples used in this study.

**Pan-cancer analysis of TAD boundaries and (super-) enhancers**. Deletion and co-duplication data were downloaded from ICGC[47]. Then, deletions and co-duplications were categorized based on their size ranging from 250 kb to 10 Mb. This data were combined with boundary strength data (from the cell lines included in this study) and the closest boundaries to each structural variant were identified using BEDTools[48]. Data for super-enhancers were downloaded from the super-enhancer archive (SEA)[37], whereas enhancer data were downloaded from FANTOM[49]. Then, the fraction of boundaries or enhancer/super-enhancer elements was normalized with the total size of the corresponding structural variation data (deletions or tandem duplications) and plotted against boundary strength. Statistical significance was assessed using paired two-sided Wilcoxon rank-sum test. The boxplots represent the distribution of values (fraction of boundaries or enhancer/super-enhancers in proximity to TAD boundary categories I through V) across the Hi-C samples used in this study.

**Code availability**. To ensure reproducibility of our analyses and make the code easily available, we incorporated all the code used to perform the work described in this study into the current version of Hi-C-bench (https://github.com/NYU-BFX/hic-bench). Updates include all the additional steps, we developed for this study: fused 2D lasso, 2D mean filter smoothing, HiCRep, calCB, optimal lambda, TAD boundary strength, and integration with ICGC copy number variation data.

**Data availability**. Detailed information about all Hi-C and ChIP-seq data sets used in this study are made available in Supplementary Data 1.

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

## Acknowledgements

We would like to thank all members of the Tsirigos and Aifantis Laboratories for critical evaluation of the manuscript, and Andreas Kloetgen who provided guidance on identifying specific interactions at fine resolutions. We would like to thank the Applied Bioinformatics Laboratories (ABL) at the NYU School of Medicine for providing bioinformatics support and helping with the analysis and interpretation of the data. This work has used computing resources at the NYU High Performance Computing Facility (HPCF). We also thank the Genome Technology Center (GTC) for expert library preparation and sequencing, and the Applied Bioinformatics Laboratories (ABL) for bioinformatics support. GTC and ABL are Shared Resources partially supported by the Cancer Center Support Grant, P30CA016087, at the Laura and Isaac Perlmutter Cancer Center. The study was supported by the American Cancer Society (RSG-15-189-01-RMC to A.T.), a Leukemia and Lymphoma Society New Idea Award (8007-17 to A.T.), and by NCI funding (R01CA169784, R01CA194923, R01CA216421 to I.A.).

## Author contributions

A.T. conceived this study. A.T., A.L. and P.K. proposed the use of lasso optimization on Hi-C contact matrices. A.T. designed the computational experiments. Y.G. extended Hi-C-bench (HiCRep, mean 2D filter smoothing and calCB). Y.G., C.L. and A.T. performed computational analyses and generated figures. T.S. performed analysis of Hi-C matrices at fine resolutions using graph-fused lasso. P.N. performed the CUTLL1 Hi-C experiments. P.N. and I.A. offered biological insights and helped with the interpretation of Hi-C data. A.T. wrote the Methods section and Results with help from C.L. C.L. wrote the Introduction and Discussion with help from A.T. All authors read and approved the final manuscript.

## Additional information

**Competing interests:** The authors declare no competing financial interests.

