## [Peer Review File · Nature Communications]

Reviewers' comments:

Reviewer #1 (Remarks to the Author):

In this paper, the authors proposed a fused two-dimensional lasso method to improve the Hi-C contact matrix reproducibility. They also characterized the TAD boundary strength, reported the relationship between TAD boundary strength and CTCF levels, and found the co-duplication of TAD boundaries and super-enhancers in cancer cells. Although this work addresses an interesting research question, there are major flaws in the proposed method and the data analysis procedure, making the results difficult to interpret. Here are my specific comments:

Major comments:

1. Page 4, line 100~101: "all datasets were down-sampled such that ...". Down-sampling the deeply sequenced Hi-C data will lead to severe loss of information, and reduce the power of TAD boundary detection. The authors need to use the original Hi-C dataset for TAD boundary analysis, and repeat their analysis and comparison. The measurement of TAD boundary strength should be robust to different sequencing depths.
2. Page 5, line 111: The authors scale Hi-C count by "effective length of the corresponding pair of interacting bins". They also need to normalize GC content and mappability score, as described in Yaffe and Tanay 2011 paper.
3. Page 5, line 123: Since Hi-C data is highly skewed, it is not convincing to use the z-score transformation. The authors can consider other asymmetric distribution which better fits real Hi-C data.
4. Page 5, "We calculated two types of correlation for Hi-C matrices". Pearson correlation coefficients between two Hi-C replicates is not an appropriate measurement of the proposed new Hi-C analysis method. The authors need to compare processed Hi-C data with other independent datasets, including deeply sequenced in situ Hi-C data (Rao et al, 2014), ChIA-PET data, capture Hi-C data and imaging. Comprehensive comparisons are necessary to fully justify the proposed fused two-dimensional lasso method.
5. In the analysis of cancer Hi-C data, the authors need to consider the effect of copy number variations, as described in Wu and Michor, *Bioinformatics* (2016) 32 (24): 3695-3701. Are their findings consistent after adjusting for CNV biases?
6. In Figure 2a, what is the bin resolution? Different λ provides very similar Hi-C contact matrices. It is more convincing to first identify significant chromatin interacting peaks from each matrices, and then compare peaks with the high resolution chromatin loops reported by Rao et al *Cell* 2014 paper.

Minor comments:

1. Page 3, line 65: "robust estimate of Hi-C contact matrices". Hi-C count data directly measure the chromatin interaction frequency. It is not clear what subject the authors want to estimate from Hi-C count matrices.
2. Page 6, line 133, "Hi-C datasets are inherently noisy". This sentence is not accurate. Hi-C data, especially, the intra-chromosomal interactions, are highly reproducible between biological replicates.

The authors need to rephrase this sentence.

3. In practice, how to choose the tuning parameter λ in the proposed fused 2D lasso method? The authors need to provide some general guidelines for practitioners.

Reviewer #2 (Remarks to the Author):

In the manuscript "Robust estimation of Hi-C contact matrices by fused lasso reveals preferential insulation of super-enhancers by strong TAD boundaries and a synergistic role in cancer", Gong and Lazaris et al. adopted the fused two-dimensional lasso method to reduce the noise level and reproducibility of Hi-C contact matrices. The penalty parameter λ in the fused lasso method penalizes differences between neighboring entries in a contact matrix. Larger penalty parameter λ allows less tolerance of neighboring differences thereby resulting in a smoother contact matrix. By categorizing TAD boundaries using different penalty settings, the authors found an association between the penalty parameter value and the TAD boundary strength. Accordingly, the authors classified TAD boundaries into five classes from the weakest boundaries to the strongest ones. Subsequent analyses showed that TAD boundary strength is associated with CTCF binding levels -- stronger CTCF binding is related to stronger boundary insulation. In addition, the authors found that super-enhancers are more likely located at TADs with at least one strong boundary. The author further claimed that strong TAD boundaries are often co-duplicated with super-enhancers and oncogenes in cancer cells.

The authors provided an interesting method for smoothing Hi-C contact matrix and improving reproducibility between Hi-C biological replicates (both same-enzyme and cross-enzyme). However, the authors didn't provide convincing assessments on the performance of the proposed method.

Major comments

1. Although the fused lasso smoothing improves reproducibility of the Hi-C contact matrices, this method would also lead signal loss in the contact matrix. In the proposed fused lasso approach, the penalty parameter λ is a trade-off between signal and noise. Larger λ could increase reproducibility but cause substantial signal loss. The authors didn't provide any discussion on what criteria should be used to select a proper penalty parameter λ in a given Hi-C dataset.
2. The authors should provide a literature review on smoothing methods for Hi-C contact matrices. Is the proposed fused lasso method the first approach to address such problem? If not, then the authors should compare their method against existing ones.
3. In the manuscript, the resolutions of Hi-C contact matrices were coarse (20 - 100 kb). The proposed smoothing method would be most useful in high-resolution settings. The authors used the Rao et al. data which provided 1 kb fine-resolution contact matrices. It will significantly strengthen the manuscript if the authors could demonstrate the benefits of their method in high-resolution settings.
4. The authors didn't provide any technical details on how the fused lasso method was implemented. In addition, no source code was supplied so the reproducibility of the work could not be assessed.
5. The authors used Pearson correlation coefficients to measure reproducibility between Hi-C contact matrices. Pearson correlation coefficient may not be the most suitable method to compare Hi-C data. It would be better if the authors could also consider other reproducibility measurements (for example,

HiC-Spector by Yan et al. and HiCRep by Yang et al.)

6. The authors mentioned that the simple insulation score (intra/inter ratio, defined on page 7) is consistent with the lasso penalty parameter for quantifying the of TAD boundary strength. In that case, what is the advantage of using the complicated fused lasso method?

7. The authors claimed that super-enhancers are co-duplicated with strong boundaries in cancer. The definition of co-duplication was not clearly explained. In cancer cells, copy number changes might also lead to false TAD identifications or inaccurate TAD boundaries strength estimations. Could the authors discuss or address that issue?

Reviewer #3 (Remarks to the Author):

In this paper by Gong et al, entitled " Robust estimation of Hi-C contact matrices by fused lasso reveals preferential insulation of super-enhancers by strong TAD boundaries and a synergistic role in cancer", the authors present a new method to estimate Hi-C contact matrixes which improve matrix reproducibility and to classify TADs based on their strength. The authors apply this method to several cell types and show that boundary strength is variable across cell types and correlates with CTCF level. Moreover, they observe that super-enhancers (SE) are preferentially located in TADs with strong boundaries and SE-strong boundaries are frequently co-duplicated in cancer.

The identification and characterization of TADs is a relevant problem since several results indicate that, although these structures seem to be conserved across different cell types, their boundaries show different strength level. In addition, the evidence of a nested hierarchy of TADs hampers the identification and characterization of biologically significant differences among samples.

Nevertheless, although addressing an extremely interesting and timely topic, I still find that the authors would need to address some significant limitations of the manuscript, which, in its present form, is not well structured, thus casting doubts on the robustness of results.

Major concerns:

1. First of all, the manuscript lacks of both methodological details and appropriate references and this hinders to fully understand the value and reproducibility of the results. For instance, the authors do not explicitly indicate the name and source of the various sample data (e.g. GSM or SRX accession), while this is necessary given that some of the studies analyzed in this work contain samples with variable quality. The authors mention in-house generated data, but do not give any reference or share this dataset. The same holds true for all other genomic data such as CTCF and acetylation.

2. Moreover, most of the Hi-C matrix "estimation" procedure builds on a previous work by the same authors (i.e., alignment, filtering and normalization is performed with HiC-bench). As such, it is not clear what innovation this work is adding to HiC-bench, rather than introducing an additional step in between normalization and TAD calling to improve reproducibility and quantify TAD boundary strength.

3. The improvement in reproducibility by itself is not convincing enough to justify the novelty since, here, the authors fail in demonstrating if forcing an improvement in reproducibility comes at the cost of losing, e.g., finer detailed features as point interactions. The authors seem to ignore that an elevated Hi-C matrix reproducibility implies, first of all, a good reproducibility of the biological samples which, given the complexity of the Hi-C protocols, may be, in these days, extremely hard to obtain. The authors should do a better job of demonstrating that i) they are not imposing reproducibility by imposing absence of variation or by flattening the signals; and ii) that they are effectively reducing the noise without losing any relevant biological information. What is the correlation among unrelated samples after the lasso procedure? Does lasso preserve cell specificity or correlation between the

same cell line from different labs? is it comparable to different cell lines from the same lab? Are we still able to capture cell specific loops?

4. The bold statement of the title "... reveals...a synergistic role in cancer" is not fully supported by the results presented through the manuscript, where only an enrichment in co-localization of SE and strong boundaries is reported.

5. Overall, the authors paid little accuracy in the statistical description of their results: some figures report symbols to indicate statistical significance (i.e., stars) but no explanation of the statistical test, of the significance threshold, etc is given. In general, figures and figure legends lack of the necessary details to understand what is plotted, the exact data source, etc. Hi-C, genomic, and genetic data are not described at all: the authors should consider expanding both data and methods description, e.g., in supplementary information, to improve clarity and understanding of their findings.

6. No code or data has been made public available neither details on the availability of the method are reported. No specification is given about the computational resources (time and memory) needed to use it.

7. Finally, there is no comparison with existing TAD callers which assign scores to boundaries, as, for instance, TADbit or Insulation score (Crane) or the "ratio" insulation score presented by the author themselves in HiC-bench. These simpler scores could provide results and correlations with super-enhancer similar to that of the 2-dimensional fused lasso procedure. This comparison is mandatory to assess the benefit of such an approach over existing methods.

Following, are some specific issues that the authors need to address to improve the clarity and the robustness of their manuscript.

METHODS section

Overall, the method section should not contain results or point to result figures.

1. The authors say they used HiC-bench but it would be nice to know some details on the pre-processing:

a. which alignment strategies? using which tool?

b. which filtering strategies? using which tool?

c. which binning? This detail is fundamental for understanding the scaling equation of page 5 (see effective length).

2. Page 5, what is the "effective length" of the pair of interacting bins? Why in the equation the two bins have different length? Did they bin at fragment level? Maybe the use of the term "bin" here is confusing. In the paper by Yaffe and Tanay, the length is the length of the restriction fragments and then restriction fragments lengths are grouped in bins for normalization. Please clarify what is intended here.

3. Page 5, line 125: why the mean is defined "expected"? is it the actual mean or an estimate of the mean?

4. Page 6, line 141: what is lambda? What is E? this equation needs to be extensively explained together with the quantification and impact of the lambda parameter on the correlation.

Supplementary Figure 2 is hard to interpret. From Figures, it seems that lambda varies from 0 to 1, but in the TAD calling results lambda can assume values >1 up to 5.

5. Page 6, line 145: Not clear at all how one-dimensional lasso would help in obtaining the full hierarchy of TADs. Moreover, the hierarchy of TADs is never mentioned in the result section.

6. Page 6, lines 148: It is hard to understand here what is represented in Supp Figure 2. This should be cited in the results section.

7. Page 7, line 160: It is not clear at all if this score has been used to call TADs or to assign an insulation score to already called TAD boundaries. In both cases, more details are needed: if it is used to call TADs, which threshold is to be set? If it is used to assign a score, does it work only for boundaries within adjacent TADs?

8. Page 7, line 175: When is this classification used? Here lambda seems to vary with a step of 0.25

- (e.g., the second category is related to $\lambda=0.2$). This doesn't appear to be the same classification used in the results section (where we have 5 classes with λ in the range 0-5).
9. Page 7, line 180 and 191: the source of the data for CTCF and H3K27ac is missing.
 10. Page 7, line 182: how was ChIP-seq data re-processed?
 11. No details are given to describe the analyzed Hi-C datasets: this is necessary to interpret at least the heatmap of Figure 3c.

RESULTS section

1. Figure 1a: distance normalization is indicated as optional. Does it impact lasso normalization?
2. Figure 1b: There appears to be no benefit of scaling over ICE or simple filtering especially at lower resolutions and for different enzymes. The author should comment on what reported in this figure panel.
3. Page 11, line 274: what is the best way to select the λ parameter?
4. Page 11, line 284: how many values of λ have been considered?
5. Page 11, line 287: what is the procedure used to identify TADs? At which resolution? Are they nested? Are they continuous? The complete list of TADs has been called with $\lambda=0$ or before lasso application? How is the list of TADs impacted by lasso? What about reproducibility of TAD calls among replicates?
6. Page 11, line 289: the sentence reported here must be fully proved, especially if a TAD hierarchy and gaps are present, showing e.g., the relationship between λ and the TAD size.
7. Page 11, line 296: The ratio score is presented here as a TAD calling method, but why is it used now i.e. after TADs have already been called? So, what is the real use of this ratio score? Moreover, the author should decide how to name this value, which is called using multiple terms throughout the paper ("ratio", "insulation score" or "boundary score" as in figure 3b).
8. Page 12, line 299 and Figure 3b: The stratification of the TAD boundaries into five categories is not clear. The text refers to number 1 to 5 in Figure 3b, but these seem the values assumed by λ . In the text, the authors state that zero corresponds to lack of boundaries, but in figure 3b zero corresponds to $\lambda=0$. So, what is reported on x-axis?
9. What does the colour scale of Figure 3c represent? If classes, then it should not be a continuous but a discrete colour bar. The legend of this figure reports about "TAD boundary insulation strength", adding even more ambiguity. I'd suggest that the authors use a more consistent terminology throughout. In addition, they should add details on the clustering method and on the heatmap. For instance, how have boundaries (rows) been ordered?
10. Page 12, line 303: The Figure could be improved indicating studies and cell type at the bottom (e.g., to show that the 4 IMR90 samples come from different studies or that K562 and KBM7 are both CML).
11. Page 12, line 311 and figure 3d: what do dots represent? Are these data from all cell lines together?
12. Page 13, lines 327: what reported here might be related to the fact that TADs with stronger domains are bigger?
13. Page 13, lines 333-341: It is not clear how figure 6a support these claims. Also, the different classes shown in figure 6a must be fully described.
14. Page 13, line 349 and Supp Figure 6b: is this figure really needed or wouldn't have been better to describe the data about genetic alterations in cancer in the Method section?
15. Page 13, line 352 and Figure 4a: How many boundaries fall in each category? I would expect fewer boundaries with strength=5 than <5. Is it really significant what is observed here? What do dots represent? Which statistical test is performed? Maybe an enrichment test (observed over expected) would be more correct than a (supposedly) t-test between groups.
16. Page 16, line 360 and Figure 4c: The authors showed above that SE are close to strong boundaries. Is it significant what we observe in figure 4c or is it what we would expect by chance simply taking a list SE? maybe SE co-duplicated with a boundary should be compared with SE co-duplicated without a

boundary and see if, in this case, the closest boundaries are weaker. Or see what happens when considering typical enhancer instead of SE.

17. Page 14, lines 366-369: the example of just one gene is a little too few to support the conclusion that amplified/over-expressed oncogenes and co-duplicated super-enhancers are, generally, close to strong TAD boundaries. Wouldn't it be possible to support this assertion with a genome-wide analysis? Also, Hi-C data are mostly from normal cells, which does not automatically imply that the same TAD structure/boundary strength/super-enhancer distribution is replicated in cancer cells. I'd be slightly more cautious on this conclusion.

Response to the reviewers' comments

We would like to thank the reviewers for their comments. We found them very detailed and extremely useful, and we believe that by addressing their concerns we have improved our manuscript. The major points we have addressed are summarized below:

- We utilized a recently published method (HiCRep) to compare Hi-C matrices, showing consistent results compared to our distance-normalized approach in the previous version of the manuscript
- We corrected Hi-C matrices for CNVs, using the published calCB method
- We compared fused lasso to another “smoothing” method, presented as part of the HiCRep paper
- We studied the trade-off between “smoothing” and preservation of cell type specificity of “smoothed” Hi-C matrices
- We demonstrated that our proposed scaling by effective length also corrects for GC content and mappability
- We applied fused lasso to matrices of finer resolutions and compared to published high-resolution loops
- We provided more details about the methods, statistical tests, packages used etc.
- We performed a robustness analysis (by sequencing depth, matrix preprocessing method, etc) of all of our results: matrix reproducibility assessment as well as association of TAD boundary strength with CTCF, super-enhancers, and CNVs
- We compared our TAD boundary calling/scoring method (“ratio”) to another widely used method (“Crane”) with respect to TAD boundary stratification by insulation score.

A detailed point-by-point response can be found below.

Reviewer #1 (Remarks to the Author):

In this paper, the authors proposed a fused two-dimensional lasso method to improve the Hi-C contact matrix reproducibility. They also characterized the TAD boundary strength, reported the relationship between TAD boundary strength and CTCF levels, and found the co-duplication of TAD boundaries and super-enhancers in cancer cells. Although this work addresses an interesting research question, there are major flaws in the proposed method and the data analysis procedure, making the results difficult to interpret. Here are my specific comments:

We have found the reviewer's comments and suggestions extremely useful, and we have carried out several additional analyses to address the concerns raised in the reviewer's critique.

Major comments:

1. Page 4, line 100~101: "all datasets were down-sampled such that ...". Down-sampling the deeply sequenced Hi-C data will lead to severe loss of information, and reduce the power of TAD boundary detection. The authors need to use the original Hi-C dataset for TAD boundary analysis, and repeat their analysis and comparison. The measurement of TAD boundary strength should be robust to different sequencing depths.

We would like to clarify that downstream analysis (TAD calling and properties of TAD boundaries) was performed at high sequencing depth. Down-sampling at different sequencing depths, was only performed for the benchmarking of our method and testing the robustness of the results. Available Hi-C datasets (*with biological replicates*) unfortunately have a wide range of useful intra-chromosomal read pairs (~34 to ~190 million read pairs, with a median of ~80 million pairs). To properly assess reproducibility across multiple samples, they all need to be sampled down to the same sequencing depth. Therefore, we report our reproducibility assessment at various sequencing depths. Importantly, as mentioned by the reviewer, we need to show that all results in the paper are "robust to different sequencing depths". We decided to extend our robustness analysis to cover not only the benchmarking of the fused 2D lasso method (**Figure 2**), but also the association of TAD boundary strength with CTCF, super-enhancers, and CNVs (**Figures 4 and 5**). The results of the robustness analysis are presented in the Supplementary Figures listed in the table below:

	High sequencing depth	Low sequencing depth
Reproducibility by enzyme and distance	Supp Figure 4	Supp Figure 3
Reproducibility assessment	Figure 2	Supp Figure 5
Association with CTCF and SEs	Figure 4	Supp Figures 8 and 9
Association with CNVs and SEs	Figure 5	Supp Figure 12

2. Page 5, line 111: The authors scale Hi-C count by "effective length of the corresponding pair of interacting bins". They also need to normalize GC content and mappability score, as described in Yaffe and Tanay 2011 paper.

Hu et al. (Hu et al., 2012) have demonstrated that the effective length is the dominant source of bias, so we expected that correcting for it would be sufficient to correct for the other two known biases. However, we agree with the reviewer that this needs to be demonstrated specifically for our scaling method. Thus, we performed a bias assessment analysis as described in (Wu and Michor, 2016) which demonstrates that our scaling method corrects for the three known enzyme-specific biases. The results are shown in **Supplementary Figure 2b**.

3. Page 5, line 123: Since Hi-C data is highly skewed, it is not convincing to use the z-score transformation. The authors can consider other asymmetric distribution which better fits real Hi-C data.

We agree that Hi-C data in general are highly skewed primarily because of the dependence of the Hi-C signal on the distance between interacting bins (**Supplementary Figure 3c**). To address this,

we apply the z-score on every diagonal (i.e. distance between loci) separately. Our approach is similar to other published methods (Yan et al., 2017). The purpose of introducing a z-score (which is applied independently for each distance) is not to fit the Hi-C data, but to model only the noise in the Hi-C data, i.e. the non-specific interactions, and not the entire distribution. Here, we assume that the noise approximately follows a (truncated) normal distribution. Ideally, we can model the Hi-C data as a mixture of two distributions, noise (normal) plus real interactions. Given the observation that the vast majority of interactions are non-specific, we can use a simpler approach, whereby we directly estimate the normal distribution corresponding to the noise. To do that, we now propose a more robust version of the z-score:

$$z_{ij} = \frac{y_{ij} - m(d)}{mad(d)}$$

where y_{ij} corresponds to the number of interactions between the bins i and j , $m(d)$ to the median number of interactions for distance $d=|j-i|$ and $mad(d)$ is the robust estimator of the standard deviation of the mean. The adjusted version of the z-score uses the median, and is more robust to outliers (which are expected to be the real interactions).

4. Page 5, "We calculated two types of correlation for Hi-C matrices". Pearson correlation coefficients between two Hi-C replicates is not an appropriate measurement of the proposed new Hi-C analysis method. The authors need to compare processed Hi-C data with other independent datasets, including deeply sequenced in situ Hi-C data (Rao et al, 2014), ChIA-PET data, capture Hi-C data and imaging. Comprehensive comparisons are necessary to fully justify the proposed fused two-dimensional lasso method.

The reviewer raises an important point about using correlation metrics to compare Hi-C matrices. Indeed, such metrics are inappropriate when used directly on non-transformed Hi-C matrices because of the strong dependence of the Hi-C signal on the distance between the interacting loci. To address this, in the original version of the manuscript, we first applied a distance normalization approach on the Hi-C matrices, similar to other proposed approaches (Yan et al., 2017). However, we agree with the reviewer that this is a very important issue, and therefore we now also use a second approach, described in (Yang et al., 2017). In their work, Yang et al., proposed the stratum-adjusted correlation coefficient (SCC) as a better approach in quantifying Hi-C matrix similarity. The results are consistent with our approach: for simplicity, in the current version of the manuscript we only show the updated reproducibility assessment results, based on the HiCRep method (**Figures 2, 3** and corresponding **Supplementary Figures**).

As far as the second part of the comment is concerned, we would like to clarify that the focus of our work is on TAD boundaries, their classification into varying levels of insulating strength, and association with (super-)enhancers and CNV events in cancer. Therefore, we had not focused on resolutions finer than 20kb. Another reason we did not focus on finer resolutions is the lack of enough Hi-C datasets, both deeply sequenced and **with availability of biological replicates**, so that we can indeed assess the reproducibility at finer resolutions in the same way we assessed coarser resolutions. In (Rao et al., 2014), only one dataset is claimed to reach 1kb resolution (and this is possible only after combining all **technical** replicates), and all loops are reported at the 5kb

resolutions. Our conclusion therefore is that we cannot include it in our main study (i.e. **Figure 2 and 3**). Despite that, we thought it would be a good idea to at least benchmark our method against the reported loops in (Rao et al., 2014), as also suggested by reviewer #2. Since we could not assess reproducibility between replicates (due to the lack of such), we followed a different approach. We asked whether we can use fused lasso to better identify the loops reported in (Rao et al., 2014), and, in addition, to test whether this is possible assuming that we had fewer reads available. In other words, we tested whether it is possible to compensate for the lack of sequenced read pairs by applying lasso. Our results seem promising and are presented in **Supplementary Figure 6a**.

5. In the analysis of cancer Hi-C data, the authors need to consider the effect of copy number variations, as described in Wu and Michor, *Bioinformatics* (2016) 32 (24): 3695-3701. Are their findings consistent after adjusting for CNV biases?

We greatly appreciate the reviewer's suggestion. We have now integrated the calCB method (Wu and Michor, 2016) into the latest version of our HiC-bench platform (<https://github.com/NYU-BFX/hic-bench>). **Figures 4 and 5** are now based on the CNV-corrected matrices. The results are consistent with our previous analyses.

6. In Figure 2a, what is the bin resolution? Different λ provides very similar Hi-C contact matrices. It is more convincing to first identify significant chromatin interacting peaks from each matrices, and then compare peaks with the high resolution chromatin loops reported by Rao et al Cell 2014 paper.

As described in our response to point #4, although fine resolutions are outside the scope of this work, we did indeed perform a comparison of the peaks identified by lasso with the ones reported in (Rao et al., 2014).

Minor comments:

1. Page 3, line 65: "robust estimate of Hi-C contact matrices". Hi-C count data directly measure the chromatin interaction frequency. It is not clear what subject the authors want to estimate from Hi-C count matrices.

We would like to thank the reviewer for pointing out this subtle point, and we have eliminated this phrase.

2. Page 6, line 133, "Hi-C datasets are inherently noisy". This sentence is not accurate. Hi-C data, especially, the intra-chromosomal interactions, are highly reproducible between biological replicates. The authors need to rephrase this sentence.

We would like to thank the reviewer for this comment. Indeed, the word “noisy” was not the best choice (we meant the well-known biases in Hi-C): we have rephrased it. We would like to also note that, despite the relatively high reproducibility, detection of specific interactions from Hi-C data based solely on Hi-C, is still challenging in the absence of high sequencing depth, as a recent study demonstrated (Forcato et al., 2017).

3. In practice, how to choose the tuning parameter λ in the proposed fused 2D lasso method? The authors need to provide some general guidelines for practitioners.

We have now proposed a method to tune parameter λ . The idea is to increase λ up to the point that we gain no further statistically significant improvement. This is now embedded as a new step in the HiC-bench platform (“optimal- λ ”), so that users of the pipeline can have the optimal λ automatically computed for each sample. The statistical test and overall process is now demonstrated in **Supplementary Figure 7c**.

Reviewer #2 (Remarks to the Author):

In the manuscript "Robust estimation of Hi-C contact matrices by fused lasso reveals preferential insulation of super-enhancers by strong TAD boundaries and a synergistic role in cancer", Gong and Lazaris et al. adopted the fused two-dimensional lasso method to reduce the noise level and reproducibility of Hi-C contact matrices. The penalty parameter λ in the fused lasso method penalizes differences between neighboring entries in a contact matrix. Larger penalty parameter λ allows less tolerance of neighboring differences thereby resulting in a smoother contact matrix. By categorizing TAD boundaries using different penalty settings, the authors found an association between the penalty parameter value and the TAD boundary strength. Accordingly, the authors classified TAD boundaries into five classes from the weakest boundaries to the strongest ones. Subsequent analyses showed that TAD boundary strength is associated with CTCF binding levels -- stronger CTCF binding is related to stronger boundary insulation. In addition, the authors found that super-enhancers are more likely located at TADs with at least one strong boundary. The author further claimed that strong TAD boundaries are often co-duplicated with super-enhancers and oncogenes in cancer cells.

The authors provided an interesting method for smoothing Hi-C contact matrix and improving reproducibility between Hi-C biological replicates (both same-enzyme and cross-enzyme). However, the authors didn't provide convincing assessments on the performance of the proposed method.

We would like to thank the reviewer for finding our work of potential interest, and for all the thoughtful comments and suggestions. We have tried to address all concerns and believe that the reviewer's comments helped us to significantly improve the quality of our work.

Major comments

1. Although the fused lasso smoothing improves reproducibility of the Hi-C contact matrices, this method would also lead signal loss in the contact matrix. In the proposed fused lasso approach, the penalty parameter λ is a trade-off between signal and noise. Larger λ could increase reproducibility but cause substantial signal loss. The authors didn't provide any discussion on what criteria should be used to select a proper penalty parameter λ in a given Hi-C dataset.

The reviewer is absolutely correct in that smoothing may lead to signal loss. One way to address this – also proposed by reviewer #3 (see major comment #3), is to compare H-C data from different samples (e.g. stem cells to differentiated cell lines) and test whether fused lasso improves matrix similarity as well as it does for matrices obtained from the same cell line. We have performed the proposed analysis, added a new section in the Results and demonstrate our findings in **Figure 3**. We show that fused lasso improves similarity significantly more in intra-vs-inter cell type Hi-C matrices, by a wide margin, particularly for our proposed preprocessing method (i.e. scaling + lasso).

Regarding tuning the lambda parameter, we have now proposed a method to tune it. The idea is to increase lambda up to the point that we gain no further statistically significant improvement. This is now embedded as a new step in the HiC-bench platform (“optimal-lambda”), so that users of the pipeline can have the optimal lambda automatically computed for each sample. The statistical test and overall process is now demonstrated in **Supplementary Figure 7c**.

2. The authors should provide a literature review on smoothing methods for Hi-C contact matrices. Is the proposed fused lasso method the first approach to address such problem? If not, then the authors should compare their method against existing ones.

We have done the literature review and found one method, very recently published as part of the HiCRep reproducibility assessment method (Yang et al., 2017). A comparison of this method with the fused lasso approach is shown in **Figure 3d**.

3. In the manuscript, the resolutions of Hi-C contact matrices were coarse (20 - 100 kb). The proposed smoothing method would be most useful in high-resolution settings. The authors used the Rao et al. data which provided 1 kb fine-resolution contact matrices. It will significantly strengthen the manuscript if the authors could demonstrate the benefits of their method in high-resolution settings.

We agree with the reviewer that addressing reproducibility of interactions at fine resolutions is an important topic of research. At the same time, we would like to clarify that the focus of our work is on TADs, their classification into varying levels of insulating strength, and association with

(super-)enhancers and CNV events in cancer. Therefore, we had not focused on resolutions finer than 20kb. Another reason we did not focus on finer resolutions is the lack of enough Hi-C datasets, both deeply sequenced and **with availability of biological replicates**, so that we can indeed assess the reproducibility at finer resolutions in the same way we assessed coarser resolutions. In (Rao et al., 2014), only one dataset is claimed to reach 1kb resolution (and this is possible only after combining all **technical** replicates), and all loops are reported at the 5kb resolutions. Our conclusion therefore is that we cannot include it in our main study (i.e. **Figure 2 and 3**). Despite that, we thought it would be a good idea to at least benchmark our method against the reported loops in (Rao et al., 2014). Since we could not assess reproducibility between replicates (due to the lack of such), we followed a different approach. We asked whether we can use fused lasso to better identify the loops reported in (Rao et al., 2014), and, in addition, to test whether this is possible assuming that we had fewer reads available. In other words, we tested whether it is possible to compensate for the lack of sequenced read pairs by applying lasso. Our results seem promising and are presented in **Supplementary Figure 6a**.

4. The authors didn't provide any technical details on how the fused lasso method was implemented. In addition, no source code was supplied so the reproducibility of the work could not be assessed.

We would like to thank the reviewer for this comment as we acknowledge that in the initial submission we did not provide sufficient details on the source code and the implementation. For fused lasso, we used the flsa R package (<https://cran.r-project.org/web/packages/flsa/index.html>). The wrapper for fused lasso (which is calling the flsa R package) is "hicseq-matrix-estimated.tcsh".

In addition, we have updated the online repository of HiC-bench platform (<https://github.com/NYU-BFX/hic-bench>) to include all the additional steps we developed for this study (fused 2D lasso, 2D mean filter smoothing, HiCRep, CalCB, etc).

5. The authors used Pearson correlation coefficients to measure reproducibility between Hi-C contact matrices. Pearson correlation coefficient may not be the most suitable method to compare Hi-C data. It would be better if the authors could also consider other reproducibility measurements (for example, HiC-Spector by Yan et al. and HiCRep by Yang et al.)

We agree with the reviewer and appreciate the suggestion to use a more appropriate method. We have incorporated HiCRep (Yang et al., 2017) into HiC-bench and replaced the Pearson correlation coefficients with stratum-adjusted correlation coefficients (SCCs) to better measure the reproducibility between Hi-C contact matrices in all of our revised figures. We would like to clarify that in the original version of the manuscript, we first applied a distance normalization approach on the Hi-C matrices, similar to other proposed approaches (Yan et al., 2017) before using the Pearson correlation to account for the dependence of Hi-C matrices on the distance

between interacting loci. However, we agree with the reviewer that this is a very important issue, and therefore we now use SCCs instead of Pearson correlation. The results are consistent with our previous findings: in the current version of the manuscript, for simplicity, we only show the updated reproducibility assessment results, based on the HiCRep method (**Figures 2, 3** and corresponding **Supplementary Figures**).

6. The authors mentioned that the simple insulation score (intra/inter ratio, defined on page 7) is consistent with the lasso penalty parameter for quantifying the of TAD boundary strength. In that case, what is the advantage of using the complicated fused lasso method?

It seems that we failed to clarify in our original submission that the boundary strength categories were not derived from the application of lasso. To clarify our approach, we now renamed the 5 boundary strength categories from 1-5 to I-V, so that they are not confused with values of lambda. **Supplementary Figure 7b** is meant to demonstrate a connection between lambda, insulation scores and make a point about the existence of a nested TAD hierarchy, but was not used for subsequent analysis. Of course, optimal lambda fused 2D lasso smoothing is still used to obtain TADs and insulation scores for the downstream analysis. To clarify these points even further, we have separated this part of the Results into two sections: "*Fused lasso reveals a TAD nested hierarchy linked to TAD boundary insulation scores*" and "*Stratification of TAD boundaries by insulating score reveals an association with enriched CTCF levels*". We also added a detailed workflow in **Figure 4a**.

7. The authors claimed that super-enhancers are co-duplicated with strong boundaries in cancer. The definition of co-duplication was not clearly explained. In cancer cells, copy number changes might also lead to false TAD identifications or inaccurate TAD boundaries strength estimations. Could the authors discuss or address that issue?

By super-enhancer/boundary co-duplication, we simply mean that a super-enhancer and a TAD boundary are found on the same segment of a tandem duplication event reported by the International Cancer Genome Consortium database (Zhang et al., 2011). Also, thanks to reviewer #1's excellent suggestion, we now use CNV-corrected Hi-C matrices, calculated using the published calCB method (Wu and Michor, 2016), and show that our results are consistent with our previous analysis (**Figures 4 and 5**).

Reviewer #3 (Remarks to the Author):

In this paper by Gong et al, entitled " Robust estimation of Hi-C contact matrices by fused lasso reveals preferential insulation of super-enhancers by strong TAD boundaries and a synergistic role in cancer", the authors present a new method to estimate Hi-C contact matrixes which

improve matrix reproducibility and to classify TADs based on their strength. The authors apply this method to several cell types and show that boundary strength is variable across cell types and correlates with CTCF level. Moreover, they observe that super-enhancers (SE) are preferentially located in TADs with strong boundaries and SE-strong boundaries are frequently co-duplicated in cancer. The identification and characterization of TADs is a relevant problem since several results indicate that, although these structures seem to be conserved across different cell types, their boundaries show different strength level. In addition, the evidence of a nested hierarchy of TADs hampers the identification and characterization of biologically significant differences among samples. Nevertheless, although addressing an extremely interesting and timely topic, I still find that the authors would need to address some significant limitations of the manuscript, which, in its present form, is not well structured, thus casting doubts on the robustness of results.

We would like to thank the reviewer for this very thorough review, we believe that the comments helped us improve the quality of our work. Our responses can be found below.

Major concerns:

1. First of all, the manuscript lacks of both methodological details and appropriate references and this hinders to fully understand the value and reproducibility of the results. For instance, the authors do not explicitly indicate the name and source of the various sample data (e.g. GSM or SRX accession), while this is necessary given that some of the studies analyzed in this work contain samples with variable quality. The authors mention in-house generated data, but do not give any reference or share this dataset. The same holds true for all other genomic data such as CTCF and acetylation.

We have now included detailed information about all datasets used in this study in **Supplementary Table 1.**

2. Moreover, most of the Hi-C matrix “estimation” procedure builds on a previous work by the same authors (i.e., alignment, filtering and normalization is performed with HiC-bench). As such, it is not clear what innovation this work is adding to HiC-bench, rather than introducing an additional step in between normalization and TAD calling to improve reproducibility and quantify TAD boundary strength.

We would like to clarify that the proposed fused lasso method, as well as the idea of TAD boundary stratification and its association with CTCF, super-enhancers and CNVs, was not part of the HiC-bench publication. In that work, we introduced HiC-bench as a platform for comprehensive Hi-C data analysis with parameter exploration capabilities. Here we propose an optimization method, we benchmark it, and then we used the optimized matrices to define 5 categories of boundary strengths and explore their properties. Importantly, to facilitate our benchmark, and ensure reproducibility of our analysis and make the code easily available, we incorporated all the code used to perform the work described in this study into the current

version of HiC-bench. Additionally, thanks to the reviewers' comments, we have incorporated additional tools into HiC-bench: HiCRep, 2D mean filter smoothing and calCB.

3. The improvement in reproducibility by itself is not convincing enough to justify the novelty since, here, the authors fail in demonstrating if forcing an improvement in reproducibility comes at the cost of losing, e.g., finer detailed features as point interactions. The authors seem to ignore that an elevated Hi-C matrix reproducibility implies, first of all, a good reproducibility of the biological samples which, given the complexity of the Hi-C protocols, may be, in these days, extremely hard to obtain. The authors should do a better job of demonstrating that i) they are not imposing reproducibility by imposing absence of variation or by flattening the signals; and ii) that they are effectively reducing the noise without losing any relevant biological information. What is the correlation among unrelated samples after the lasso procedure? Does lasso preserve cell specificity or correlation between the same cell line from different labs? is it comparable to different cell lines from the same lab? Are we still able to capture cell specific loops?

The reviewer is absolutely correct in that smoothing may lead to signal loss. One way to address this, is to compare unrelated samples (e.g. stem cells to differentiated cell lines) and test whether fused lasso improves matrix similarity as well as it does for matrices obtained from the same cell line. We have performed the proposed analysis and demonstrate the results in **Figure 3**. We show that fused lasso improves similarity significantly more in intra-vs-inter cell type Hi-C matrices, and by a wide margin, particularly for our proposed preprocessing method (i.e. scaling + lasso).

As far as the second part of the comment is concerned, we would like to clarify that the focus of our work is on TAD boundaries, their classification into varying levels of insulating strength, and association with (super-)enhancers and CNV events in cancer. Therefore, we had not focused on resolutions finer than 20kb. Another reason we did not focus on finer resolutions is the lack of enough Hi-C datasets, both deeply sequenced and **with availability of biological replicates**, so that we can indeed assess the reproducibility at finer resolutions in the same way we assessed coarser resolutions. In (Rao et al., 2014), only one dataset is claimed to reach 1kb resolution (and this is possible only after combining all **technical** replicates), and all loops are reported at the 5kb resolutions. Our conclusion therefore is that we cannot include it in our main study (i.e. **Figure 2 and 3**). Despite that, we thought it would be a good idea to at least benchmark our method against the reported loops in (Rao et al., 2014), as also suggested by reviewer #2. Since we could not assess reproducibility between replicates (due to the lack of such), we followed a different approach. We asked whether we can use fused lasso to better identify the loops reported in (Rao et al., 2014), and, in addition, to test whether this is possible assuming that we had fewer reads available. In other words, we tested whether it is possible to compensate for the lack of sequenced read pairs by applying lasso. Our results seem promising and are presented in **Supplementary Figure 6a**.

4. The bold statement of the title “... reveals...a synergistic role in cancer” is not fully supported by the results presented through the manuscript, where only an enrichment in co-localization of SE and strong boundaries is reported.

We agree with the reviewer that a more accurate title is necessary. The new title is “*Stratification of TAD boundaries identified in reproducible Hi-C contact matrices reveals preferential insulation of super-enhancers by strong boundaries*”.

5. Overall, the authors paid little accuracy in the statistical description of their results: some figures report symbols to indicate statistical significance (i.e., stars) but no explanation of the statistical test, of the significance threshold, etc is given. In general, figures and figure legends lack of the necessary details to understand what is plotted, the exact data source, etc. Hi-C, genomic, and genetic data are not described at all: the authors should consider expanding both data and methods description, e.g., in supplementary information, to improve clarity and understanding of their findings.

We have now included detailed information on all figures. We have also expanded the methods section. Data sources are presented in detail in **Supplementary Table 1**.

6. No code or data has been made public available neither details on the availability of the method are reported. No specification is given about the computational resources (time and memory) needed to use it.

We would like to thank the reviewer for this comment as we acknowledge that in the initial submission we did not provide sufficient details on the source code and the implementation. For fused lasso, we used the flsa R package (<https://cran.r-project.org/web/packages/flsa/index.html>). The wrapper for fused lasso (which is calling the flsa R package) is “hicseq-matrix-estimated.tcsh”.

In addition, we have updated the online repository of HiC-bench platform (<https://github.com/NYU-BFX/hic-bench>) to include all the additional steps we developed for this study (fused 2D lasso, 2D mean filter smoothing, HiCRep, CalCB, etc).

Finally, in **Supplementary Figure 6b,c**, we provide information on the computational resources required by the lasso algorithm, as a function of the chromosome size.

7. Finally, there is no comparison with existing TAD callers which assign scores to boundaries, as, for instance, TADbit or Insulation score (Crane) or the “ratio” insulation score presented by the author themselves in HiC-bench. These simpler scores could provide results and correlations with super-enhancer similar to that of the 2-dimensional fused lasso procedure. This comparison is mandatory to assess the benefit of such an approach over existing methods.

It seems that we failed to clarify in our original submission that the boundary strength categories were not derived from the application of lasso. To clarify our approach, we now renamed the 5 boundary strength categories from 1-5 to I-V, so that they are not confused with values of lambda. **Supplementary Figure 7b** is meant to demonstrate a connection between lambda, insulation scores and make a point about the existence of a nested TAD hierarchy, but was not used for subsequent analysis. Of course, optimal lambda fused 2D lasso smoothing is still used to obtain TADs and insulation scores for the downstream analysis. To clarify these points even further, we have separated this part of the Results into two sections: *“Fused lasso reveals a TAD nested hierarchy linked to TAD boundary insulation scores”* and *“Stratification of TAD boundaries by insulating score reveals an association with enriched CTCF levels”*. We also added a detailed workflow in **Figure 4a**. The classification of boundaries into five categories was indeed performed using the “ratio” insulation score, applying on the optimized matrices obtained by fused lasso. As suggested by the reviewer, we now performed a comparison of boundaries categories I-V obtained by “ratio” to those obtained by “crane”. The results show that ratio-obtained boundary categories have corresponding CTCF levels that are better associated with insulating strength (**Figure 4d** and corresponding **Supplementary Figures**).

Following, are some specific issues that the authors need to address to improve the clarity and the robustness of their manuscript.

METHODS section

Overall, the method section should not contain results or point to result figures.

- 1. The authors say they used HiC-bench but it would be nice to know some details on the pre-processing:**
 - a. which alignment strategies? using which tool?**
 - b. which filtering strategies? using which tool?**
 - c. which binning? This detail is fundamental for understanding the scaling equation of page 5 (see effective length).**

It is indeed a good idea to add details of the basic Hi-C processing approach and we have now updated the corresponding methods section titled “Initial processing of published high-resolution Hi-C datasets”.

- 2. Page 5, what is the “effective length” of the pair of interacting bins? Why in the equation the two bins have different length? Did they bin at fragment level? Maybe the use of the term “bin” here is confusing. In the paper by Yaffe and Tanay, the length is the length of the restriction fragments and then restriction fragments lengths are grouped in bins for normalization. Please clarify what is intended here.**

Indeed, all bins have the same length at a fixed resolution. We do not perform fragment-level binning. However, the *effective* length, i.e. the total length of the regions that are within a certain distance (typically 500nt) from the enzyme cutting sites that correspond to each bin, is different for every bin (because each bin may have a different number of enzyme cutting sites). Reads that map outside these regions are typically discarded in Hi-C analyses. Several studies have observed biases related to the effective length, because, as expected, the smaller the effective length of a fixed-size bin, the lower the probability of reads aligning to that bin. See also **Supplementary Figure 2** and the corresponding methods section.

3. Page 5, line 125: why the mean is defined “expected”? is it the actual mean or an estimate of the mean?

This was supposed to be the actual mean. We removed the word “expected”.

4. Page 6, line 141: what is lambda? What is E? this equation needs to be extensively explained together with the quantification and impact of the lambda parameter on the correlation. Supplementary Figure 2 is hard to interpret. From Figures, it seems that lambda varies from 0 to 1, but in the TAD calling results lambda can assume values >1 up to 5.

We have now explained what E is. Lambda is the parameter of the model. The reproducibility analyses were performed with λ values up to 1.0, because no additional benefit on the correlation was observed beyond $\lambda=1.0$. In a completely separate analysis, when we performed TAD calling later in the paper, we observed that if we kept increasing λ (the reproducibility is still high), we obtained fewer and fewer TADs in a nested hierarchy (see updates about nested hierarchy in the text and our response to point #5 below).

5. Page 6, line 145: Not clear at all how one-dimensional lasso would help in obtaining the full hierarchy of TADs. Moreover, the hierarchy of TADs is never mentioned in the result section.

The hierarchy is discussed in the section “Fused lasso reveals a TAD hierarchy linked to TAD boundary insulation scores”. We have updated this section, see also our response to comment #6 (RESULTS section) below.

6. Page 6, lines 148: It is hard to understand here what is represented in Supp Figure 2. This should be cited in the results section.

This figure has now been removed as we do not use fused 1-dimensional lasso anymore (we replaced it with 2D lasso).

7. Page 7, line 160: It is not clear at all if this score has been used to call TADs or to assign an

insulation score to already called TAD boundaries. In both cases, more details are needed: if it is used to call TADs, which threshold is to be set? If it is used to assign a score, does it work only for boundaries within adjacent TADs?

We totally agree that this has not been clearly explained. We now describe TAD calling and insulation score calculations in detail in methods section “TAD calling using the ratio insulation score”.

8. Page 7, line 175: When is this classification used? Here lambda seems to vary with a step of 0.25 (e.g., the second category is related to lambda=0.2). This doesn't appear to be the same classification used in the results section (where we have 5 classes with lambda in the range 0-5).

Indeed, the two classifications are different, and from the reviewer's comment we realized that this had not been explained properly. We now present the two classifications in two separate sections in the Results to make this clearer. The λ -based classification is discussed in the section titled “Fused lasso reveals a TAD hierarchy linked to TAD boundary insulation scores”, while the TAD boundary stratification using insulation scores is discussed in the section titled “Stratification of TAD boundaries by insulating score reveals an association with enrichment in CTCF levels”. We have also described the two separate procedure in the Methods section.

9. Page 7, line 180 and 191: the source of the data for CTCF and H3K27ac is missing.

We have now included detailed information about all datasets used in this study in **Supplementary Table 1**.

10. Page 7, line 182: how was CHIP-seq data re-processed?

We added a new Methods section to describe CTCF and H3K27ac CHIP-seq data processing. Raw sequencing files were aligned using bowtie2 version 2.3.1 with standard parameters. Only uniquely mapped reads were selected for downstream analysis. PCR duplicates were removed using Picad-tools version 1.88. Macs2 version 2.0.10.20131216 were used to call narrow peaks for CTCF and broad peaks for H3K27ac with other parameters as default.

11. No details are given to describe the analyzed Hi-C datasets: this is necessary to interpret at least the heatmap of Figure 3c.

We have now included detailed information about all datasets used in this study in **Supplementary Table 1** and added a reference to this table in the legend of this figure (now **Figure 4**).

RESULTS section

1. Figure 1a: distance normalization is indicated as optional. Does it impact lasso normalization?

Distance normalization is meant to be used only for the identification of specific interactions at fine resolutions. Since this is not the main focus of our work, we removed it from the workflow of **Figure 1**, and mention it at the section where we discuss fine resolutions in response to the reviewers' comments.

2. Figure 1b: There appears to be no benefit of scaling over ICE or simple filtering especially at lower resolutions and for different enzymes. The author should comment on what reported in this figure panel.

There is indeed no benefit of scaling over ICE, especially at lower resolutions. The main point of this figure (now moved to **Supplementary Figures 3 and 4**) was to introduce the problem: exactly as the reviewer points out, at lower resolutions all preprocessing methods perform poorly. This, in our opinion, introduces the need for an optimization method, such as lasso.

3. Page 11, line 274: what is the best way to select the lambda parameter?

We now propose a method to select parameter lambda. The idea is to increase lambda up to the point that we gain no further statistically significant improvement. This is now embedded as a new step in the HiC-bench platform ("optimal-lambda"), so that users of the pipeline can have the optimal lambda automatically computed for each sample. The statistical test and overall process is now demonstrated in **Supplementary Figure 7c**.

4. Page 11, line 284: how many values of lambda have been considered?

We ranged lambda from 0 to 5 with a step of 0.25, i.e. 21 lambda values (included zero).

5. Page 11, line 287: what is the procedure used to identify TADs? At which resolution? Are they nested? Are they continuous? The complete list of TADs has been called with lambda=0 or before lasso application? How is the list of TADs impacted by lasso? What about reproducibility of TAD calls among replicates?

We have now explained TAD calling and insulation score calculations in detail in methods section "TAD calling using the ratio insulation score". Regarding TAD boundary reproducibility between replicates, we have extensively studied this in our previous work (Lazaris et al., 2017), where we showed that the "ratio" and "crane" TAD callers outperform other approaches.

6. Page 11, line 289: the sentence reported here must be fully proved, especially if a TAD hierarchy and gaps are present, showing e.g., the relationship between lambda and the TAD size.

We have now included more data showing that the TAD boundaries obtained at lower λ values are nested within the boundaries obtained at higher lambda values. Indeed, when comparing TAD boundaries detected at successive λ values, we found that higher λ values produced TAD boundaries that are almost a strict subset of TAD boundaries produced at lower λ values: we measured a ~94% overlap of TAD boundaries at a strict accuracy of considering only the exact bin as a true common boundary between matrices produced using different λ values, and a ~98% overlap when TAD boundaries are allowed to differ by at most one bin between TADs generated for successive λ values.

7. Page 11, line 296: The ratio score is presented here as a TAD calling method, but why is it used now i.e. after TADs have already been called? So, what is the real use of this ratio score? Moreover, the author should decide how to name this value, which is called using multiple terms throughout the paper (“ratio”, “insulation score” or “boundary score” as in figure 3b).

The ratio score is used both to quantify insulating strength for each 40kb bin in the genome and to identify TAD boundaries, as described in (Lazaris et al., 2017), TAD boundaries are identified as local maxima of the genome-wide ratio score. We have added a new Methods section to describe our approach in detail: “*TAD calling using the “ratio” insulation score*”. Also, we have now updated the text and unified the different terms into one (“insulation score”).

8. Page 12, line 299 and Figure 3b: The stratification of the TAD boundaries into five categories is not clear. The text refers to number 1 to 5 in Figure 3b, but these seem the values assumed by lambda. In the text, the authors state that zero corresponds to lack of boundaries, but in figure 3b zero corresponds to lambda=0. So, what is reported on x-axis?

We agree with the reviewer’s comment that number 1 to 5 in the figures are confusing, because they could correspond either to lambda values or to the five categories. We have changed the labels of the categories from arabic numbers (1-5) to latin numbers (I-V) in all of our figures to represent five different categories of boundaries with strength from low to high and avoid the confusion with lambda values. See also Methods section “*Categorization of TAD boundaries based on insulation scores*”.

9. What does the colour scale of Figure 3c represent? If classes, then it should not be a continuous but a discrete colour bar. The legend of this figure reports about “TAD boundary insulation strength”, adding even more ambiguity. I’d suggest that the authors use a more

consistent terminology throughout. In addition, they should add details on the clustering method and on the heatmap. For instance, how have boundaries (rows) been ordered?

We have now used a discrete color bar as suggested by the reviewer, updated the text and unified the different terms into one (“insulation score”).

10. Page 12, line 303: The Figure could be improved indicating studies and cell type at the bottom (e.g., to show that the 4 IMR90 samples come from different studies or that K562 and KBM7 are both CML).

We have now included detailed information about all datasets used in this study, including cell type, in **Supplementary Table 1** and added a reference to this table in the legend of this figure (now **Figure 4**).

11. Page 12, line 311 and figure 3d: what do dots represent? Are these data from all cell lines together?

Yes, these boxplots represent distributions across cell lines.

12. Page 13, lines 327: what reported here might be related to the fact that TADs with stronger domains are bigger?

This is a very interesting idea: we tested the association between the strength of each pair of consecutive TAD boundaries with the size of the TAD they define. For simplicity (to reduce the number of combinations), we defined TAD boundaries as weak when they belong to categories I or II, and strong when they belong to IV or V. Then, we calculated the TAD sizes of TADs defined by: (i) strong-strong, (ii) strong-weak, and (iii) weak-weak TAD boundaries. Here, we report the quartiles of the distributions of the TAD sizes in each scenario, perhaps surprisingly, showing that the weak-weak combination flanks (slightly) larger TADs:

- strong-strong: [520000, 640000, 880000]
- strong-weak: [520000, 720000, 1240000]
- weak-weak: [520000, 760000, 1360000]

13. Page 13, lines 333-341: It is not clear how figure 6a support these claims. Also, the different classes shown in figure 6a must be fully described.

We realized that **Supplementary Figure 6a** not only disrupts the flow, but also does not really contribute to the conclusions, so we decided to remove it altogether.

14. Page 13, line 349 and Supp Figure 6b: is this figure really needed or wouldn't have been better to describe the data about genetic alterations in cancer in the Method section?

Whereas the figure could be also included in the Methods section, we believe it serves its purpose better in the Results section, as it helps the reader appreciate the extent of deletions and tandem duplications in ICGC and how these alterations are linked to boundaries of variable strength (shown in other figures of the results section).

15. Page 13, line 352 and Figure 4a: How many boundaries fall in each category? I would expect fewer boundaries with strength=5 than <5. Is it really significant what is observed here? What do dots represent? Which statistical test is performed? Maybe an enrichment test (observed over expected) would be more correct than a (supposedly) t-test between groups.

We have ensured that we consider equal number of boundaries from each category (I to V) so that we do not introduce any bias by having unequal number of boundaries per category (see Methods "Categorization of TAD boundaries based on insulation scores"). The differences that we observe across categories in what is now **Figure 5a** are significant. We used Wilcoxon paired rank-sum test to assess significance with cut-off p-value<0.05. Instead of reporting raw numbers of boundaries or enhancer/SE elements, we now report fractions of those normalized by the total size (in Gbp) of copy-number variants, which should make the results directly comparable **across Figure 5a,b**. The dots in the boxplots represent boundaries of corresponding boundary strength category for the 26 samples for which Hi-C data and CTCF data were available.

16. Page 16, line 360 and Figure 4c: The authors showed above that SE are close to strong boundaries. Is it significant what we observe in figure 4c or is it what we would expect by chance simply taking a list SE? maybe SE co-duplicated with a boundary should be compared with SE co-duplicated without a boundary and see if, in this case, the closest boundaries are weaker. Or see what happens when considering typical enhancer instead of SE.

We would like to thank the reviewer for this very insightful comment and we appreciate the suggestion of using typical enhancers as a control. We indeed downloaded regular enhancer data from the FANTOM database and performed the same analysis on the regular enhancers. We observed that despite a slight trend for regular enhancers to be located close to strong boundaries, this trend is more significant in the case of super-enhancers (**Figure 5c** of the revised manuscript). The dots in the boxplots represent boundaries of corresponding boundary strength category for the 26 samples for which Hi-C data and H3K27ac data were available. We used Wilcoxon paired rank-sum test to assess significance with cut-off p-value<0.05.

17. Page 14, lines 366-369: the example of just one gene is a little too few to support the conclusion that amplified/over-expressed oncogenes and co-duplicated super-enhancers are,

generally, close to strong TAD boundaries. Wouldn't it be possible to support this assertion with a genome-wide analysis? Also, Hi-C data are mostly from normal cells, which does not automatically imply that the same TAD structure/boundary strength/super-enhancer distribution is replicated in cancer cells. I'd be slightly more cautious on this conclusion.

We absolutely agree with the reviewer that one gene is definitely not enough to support a connection between oncogenes, super-enhancers and TAD boundaries. We have completely removed this claim from the text. As a side note, we have indeed performed a genome-wide analysis of genes in the proximity of strong boundaries, and we found that these genes are enriched in MYC, MAZ, ETS and other interesting factors. We believe this is of potential interest, but would like to investigate it further, with more data, in future work.

Bibliography

Forcato, M., Nicoletti, C., Pal, K., Livi, C.M., Ferrari, F., and Bicciato, S. (2017). Comparison of computational methods for Hi-C data analysis. *Nat Methods* 14, 679–685.

Lazaris, C., Kelly, S., Ntziachristos, P., Aifantis, I., and Tsirigos, A. (2017). HiC-bench: comprehensive and reproducible Hi-C data analysis designed for parameter exploration and benchmarking. *BMC Genomics* 18, 22.

Wu, H.-J., and Michor, F. (2016). A computational strategy to adjust for copy number in tumor Hi-C data. *Bioinformatics* 32, 3695–3701.

Yang, T., Zhang, F., Yardimci, G.G., Song, F., Hardison, R.C., Noble, W.S., Yue, F., and Li, Q. (2017). HiCRep: assessing the reproducibility of Hi-C data using a stratum-adjusted correlation coefficient. *Genome Res*.

Zhang, J., Baran, J., Cros, A., Guberman, J.M., Haider, S., Hsu, J., Liang, Y., Rivkin, E., Wang, J., Whitty, B., et al. (2011). International Cancer Genome Consortium Data Portal—a one-stop shop for cancer genomics data. *Database (Oxford)* 2011, bar026.

Reviewers' comments:

Reviewer #1 (Remarks to the Author):

I appreciate the authors' great efforts in the revised manuscript. They have provided comprehensive evidence and detailed explanation, and significantly improved the quality of this paper. All my previous comments have been fully addressed. I don't have further comments and suggestions.

Reviewer #2 (Remarks to the Author):

In the revised manuscript, the authors have made significant improvements of their work.

- The authors provided more comprehensive evaluation and assessments of their proposed Hi-C normalization, smoothing, and TAD identification methods. In particular, the authors compared their fused lasso smoothing method to a recently published HiCRep approach.
- The authors added more technical details about the implementations of Hi-C data processing and fused lasso smoothing. In addition, the authors provided the analysis source code in their HiC-bench package.
- The authors studied the trade-off between "smoothing" and preservation of cell type specificity of "smoothed" Hi-C matrices.
- In addition to TAD boundary calling at 40kb resolution, the authors also applied fused lasso at 5kb resolution to identify fine-scale chromatin loops and showed that their rankings of significant interactions are consistent with the published chromatin loops by Rao et al. 2014.

Overall, the authors have satisfactorily addressed most of my previous comments. I have few remaining minor comments.

1. To study the trade-off between "smoothing" and information/signal loss, the authors compared the reproducibility between biological replicates (intra-cell-type) as well as the stratum-adjusted correlation coefficients (SCC) between different cell types (inter-cell-type). In addition, the authors proposed the selection of optimal λ as the smallest value with no statistically significant improvement on reproducibility. Should the author also take cell-type specificity into consideration when identifying the optimal λ ?

2. In Figure S6a, the authors reported that by tuning the λ parameter, the fused lasso method could recover 90% of the chromatin loops at 5kb resolution as reported in Rao et al. 2014. Could the authors compare the "optimal" λ in Figure S6a to the "optimal" λ values defined using the reproducibility statistical test in Figure S7c?

Reviewer #3 (Remarks to the Author):

I thank the authors for addressing most of my concerns. I believe the results are more robust and the manuscript is improved. I suggest some adjustments in order to make this work clearer and easily readable, especially in the methods section.

METHODS

The methods section is now certainly more complete and accurate. However, given the complexity and multitude of analyses, it could result confusing to the reader.

- The authors successfully explained in the response that subsampled matrixes were used mainly for

the benchmarking part of the paper and that for the downstream analysis, hic matrixes with the entire available number of reads were used. However, I am not sure that this emerges from the paper. Moreover, at what resolution were the matrixes binned for the downstream analysis?

- Page 5 line 104: there is a repetition of the filters that were applied to the reads, that had just been stated in the previous page. Moreover, there are references to colors that are obscure without a reference to supplementary figure 1A. I suggest the authors remove the repetitions and move the filter and colors explanation in the legend of the figure.
- Page 5 line 121: I suggest adding a very brief definition of "effective" length.
- Page 6 line 150: in the description of the fused 2 D lasso, the authors state that lasso is very well-suited for "identifying" topological domains. I think that this phrase is misleading since it appears that lasso is a method to call TADs instead that an optimization method.
- Page 7 line 175: there is no description on how ICE and calCB were applied.
- Page 9 line 215: there is a reference to an "optimal lambda" but the definition is given only in the results section. The choice of the optimal lambda is an important point and should be reported in the methods section.
- I suggest the authors add a code availability section, where they can report that lasso and the other tools are implemented in the updated version of hicbench, as they wrote in their reponse.

RESULTS

- What are the colors in supplementary figure 1C?
- Page 14 line 354. This additional analysis doesn't prove that lasso can "improve" detection of specific DNA-DNA interactions. The authors should consider rephrasing this part.
- Figure 4a contains a reference to figure 3c which appears to be incorrect.
- The legend of figure 2 refers to top and bottom panels which are not present in the figure.

Response to the reviewers' comments

Reviewer #1 (Remarks to the Author):

I appreciate the authors' great efforts in the revised manuscript. They have provided comprehensive evidence and detailed explanation, and significantly improved the quality of this paper. All my previous comments have been fully addressed. I don't have further comments and suggestions.

Reviewer #2 (Remarks to the Author):

In the revised manuscript, the authors have made significant improvements of their work.

- The authors provided more comprehensive evaluation and assessments of their proposed Hi-C normalization, smoothing, and TAD identification methods. In particular, the authors compared their fused lasso smoothing method to a recently published HiCRep approach.
- The authors added more technical details about the implementations of Hi-C data processing and fused lasso smoothing. In addition, the authors provided the analysis source code in their HiC-bench package.
- The authors studied the trade-off between “smoothing” and preservation of cell type specificity of “smoothed” Hi-C matrices.
- In addition to TAD boundary calling at 40kb resolution, the authors also applied fused lasso at 5kb resolution to identify fine-scale chromatin loops and showed that their rankings of significant interactions are consistent with the published chromatin loops by Rao et al. 2014.

Overall, the authors have satisfactorily addressed most of my previous comments. I have few remaining minor comments.

1. To study the trade-off between “smoothing” and information/signal loss, the authors compared the reproducibility between biological replicates (intra-cell-type) as well as the stratum-adjusted correlation coefficients (SCC) between different cell types (inter-cell-type). In addition, the authors proposed the selection of optimal λ as the smallest value with no statistically significant improvement on reproducibility. Should the author also take cell-type specificity into consideration when identifying the optimal λ ?

This is a great suggestion. We now show results using both approaches: (a) comparing SCC between replicates (as done previously; Supp Figure 7c), or (b) comparing the difference between intra- and inter-cell-type SCC (new figure panel; Supp Figure 7d). As before, optimal lambda is selected as the smallest value with no statistically significant improvement in either (a) or (b).

2. In Figure S6a, the authors reported that by tuning the λ parameter, the fused lasso method could recover 90% of the chromatin loops at 5kb resolution as reported in Rao et al. 2014. Could the authors compare the “optimal” λ in Figure S6a to the “optimal” λ values defined using the reproducibility statistical test in Figure S7c?

This is also a great suggestion, however, due to the lack of biological replicates in this particular ultra-high sequencing depth dataset, we are not able to select the optimal lambda, as our method requires at least two biological replicates. We now specify this requirement in the methods section.

Reviewer #3 (Remarks to the Author):

I thank the authors for addressing most of my concerns. I believe the results are more robust and the manuscript is improved. I suggest some adjustments in order to make this work clearer and easily readable, especially in the methods section.

METHODS

The methods section is now certainly more complete and accurate. However, given the complexity and multitude of analyses, it could result confusing to the reader.

- The authors successfully explained in the response that subsampled matrixes were used mainly for the benchmarking part of the paper and that for the downstream analysis, hic matrixes with the entire available number of reads were used. However, I am not sure that this emerges from the paper. Moreover, at what resolution were the matrixes binned for the downstream analysis?

For all downstream analyses, a bin size of 40kb was used. We now added a sentence to clarify (line 402 and footnote).

Regarding the sequencing depth, our approach had two objectives. First, to obtain results (both benchmarking and all the findings) at lower sequencing depth (i.e. 40 million usable read pairs, which corresponds to >100 million sequenced read pairs). We would like to emphasize here that this sequencing depth has been demonstrated to be sufficient for reproducible TAD calling (HiC-bench, Lazaris et al. 2017). Second, to obtain results at a higher sequencing depth of 80 million

usable read pairs (i.e. >160 million sequenced read pairs), which represents the median number of usable read pairs in our dataset. We have now added clarifications in the titles of all relevant figure legends, inside parentheses, to indicate what sequencing depth was used. Of course, the main results, presented as main figures in this manuscript, are based on the high sequencing depth. A more detailed response to this question was included as a response to a comment from Reviewer #1, which we copy below:

*We would like to clarify that downstream analysis (TAD calling and properties of TAD boundaries) was performed at high sequencing depth. Down-sampling at different sequencing depths, was only performed for the **benchmarking of our method and testing the robustness of the results.** Available Hi-C datasets (**with biological replicates**) unfortunately have a wide range of useful intra-chromosomal read pairs (~34 to ~190 million read pairs, with a median of ~80 million pairs). To properly assess reproducibility across multiple samples, they all need to be sampled down to the same sequencing depth. Therefore, we report our reproducibility assessment at various sequencing depths. Importantly, as mentioned by the reviewer, we need to show that all results in the paper are “robust to different sequencing depths”. We decided to extend our robustness analysis to cover not only the benchmarking of the fused 2D lasso method (**Figure 2**), but also the association of TAD boundary strength with CTCF, super-enhancers, and CNVs (**Figures 4 and 5**). The results of the robustness analysis are presented in the Supplementary Figures listed in the table below:*

	High sequencing depth	Low sequencing depth
Reproducibility by enzyme and distance	Supp Figure 4	Supp Figure 3
Reproducibility assessment	Figure 2	Supp Figure 5
Association with CTCF and SEs	Figure 4	Supp Figures 8 and 9
Association with CNVs and SEs	Figure 5	Supp Figure 12

- Page 5 line 104: there is a repetition of the filters that were applied to the reads, that had just been stated in the previous page. Moreover, there are references to colors that are obscure without a reference to supplementary figure 1A. I suggest the authors remove the repetitions and move the filter and colors explanation in the legend of the figure.

We have now removed the repetitive text and added the filtering and color information in the figure legend.

- Page 5 line 121: I suggest adding a very brief definition of “effective” length.

This is a great suggestion: we added a brief definition, lines 114-116.

- Page 6 line 150: in the description of the fused 2 D lasso, the authors state that lasso is very well-suited for “identifying” topological domains. I think that this phrase is misleading since it appears that lasso is a method to call TADs instead that an optimization method.

This is a good point, we have removed this sentence.

- Page 7 line 175: there is no description on how ICE and calCB were applied.

We have now added a clarification, lines 165-167.

- Page 9 line 215: there is a reference to an “optimal lambda” but the definition is given only in the results section. The choice of the optimal lambda is an important point and should be reported in the methods section.

We have added a new methods section on the selection of optimal lambda, lines 208-216.

- I suggest the authors add a code availability section, where they can report that lasso and the other tools are implemented in the updated version of hicbench, as they wrote in their reponse.

We have now added a code availability section, lines 269-274.

RESULTS

- What are the colors in supplementary figure 1C?

We added a clarification in the legend: colors represent different replicates of the IMR90 samples.

- Page 14 line 354. This additional analysis doesn’t prove that lasso can “improve” detection of specific DNA-DNA interactions. The authors should consider rephrasing this part.

We removed this phrase.

- Figure 4a contains a reference to figure 3c which appears to be incorrect.

We have removed this incorrect reference.

- **The legend of figure 2 refers to top and bottom panels which are not present in the figure.**

We removed the top/bottom reference.

REVIEWERS' COMMENTS:

Reviewer #2 (Remarks to the Author):

The authors have adequately addressed all my previous comments and questions.

Reviewer #3 (Remarks to the Author):

My requests have been addressed, I don't have further comments.